# Discovering Symbolic Partial Differential Equation by Abductive Learning

**En-Hao Gao, Cunjing Ge, Yuan Jiang, Zhi-Hua Zhou**

National Key Laboratory for Novel Software Technology, Nanjing University, China

School of Artificial Intelligence, Nanjing University, China

gaoeh@lamda.nju.edu.cn, {gecunjing, jiangyuan, zhouzh}@nju.edu.cn

## Abstract

Discovering symbolic Partial Differential Equation (PDE) from data is one of the most promising directions of modern scientific discovery. Effectively constructing an expressive yet concise hypothesis space and accurately evaluating expression values, however, remain challenging due to the exponential explosion with the spatial dimension and the noise in the measurements. To address these challenges, we propose the ABL-PDE approach that employs the Abductive Learning (ABL) framework to discover symbolic PDEs. By introducing a First-Order Logic (FOL) knowledge base, ABL-PDE can represent various PDEs, significantly constraining the hypothesis space without sacrificing expressive power, while also facilitating the incorporation of problem-specific knowledge. The proposed consistency optimization process establishes a synergistic interaction between the knowledge base and the neural network learning module, achieving robust structure identification, accurate coefficient estimation, and enhanced stability against hyperparameter variation. Experimental results on three benchmarks across different noise levels demonstrate the effectiveness of our approach in PDE discovery.

## 1 Introduction

Partial Differential Equations (PDEs) serve as fundamental mathematical tools for describing a vast array of physical phenomena in science and engineering. They effectively capture the intricate relationships between how quantities change over space and/or evolve over time. Complemented by both analytical techniques and modern numerical solvers, PDEs provide interpretable models that support reliable prediction and control. Due to their versatility, PDEs are widely employed across a range of applications, including airfoil design in aerodynamics [14], weather prediction in meteorology [9], and pricing analysis in quantitative finance [6].

Recently, physics-informed machine learning has demonstrated remarkable success in approximating the behavior of complex physical systems [3, 22], which significantly changes the way we represent and leverage PDEs. Note that a single PDE can be adapted to various scenarios by simply modifying its boundary conditions and domain geometries; however, PDEs in explicit form remain essential in applications where high reliability is paramount. Therefore, discovering symbolic governing equations from data becomes one of the most attractive research directions in scientific discovery [4].

Significant efforts [7, 10, 12, 16, 18] have been made to the PDE discovery task. Most of these methods consist of two primary components: a derivative-integral calculator and a candidate term library, as illustrated in Figure 1. The calculator evaluates symbolic expressions based on either direct derivative estimation or weak formulation-based integral computation. The library constructs the hypothesis space for the target equation, with each term representing a function of the variables of interest. The underlying PDE is typically assumed to be a linear combination of these terms and is identified through symbolic regression techniques.

39th Conference on Neural Information Processing Systems (NeurIPS 2025).

A grand challenge of PDE discovery lies in constructing a comprehensive candidate term library. On one hand, this library must be sufficiently expressive to capture patterns within data; On the other hand, it should remain compact to enable an efficient PDE structure identification. Existing studies attempt to achieve the trade-off by constraining the expressiveness using a concise library [11, 16], or emphasizing the expressiveness at the expense of using

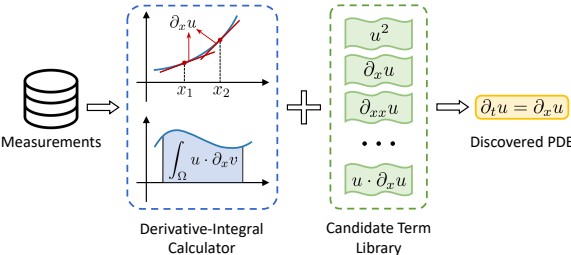

Figure 1: An illustration of PDE discovery framework.

a redundant library [8, 26, 27]. Accurately evaluating the expression values is also challenging. Because we can only access discrete measurements, derivatives or integrals have to be estimated using approximate methods, the accuracy of which can be significantly affected if there is no sufficient data or there is noise. Many studies attempt to address this issue using techniques such as ensemble learning [12] and weak formulation [11], while this paper attempts to exploit the discovered physical information to help enhance the calculator's capability, which has rarely been exploited before.

In this paper, we propose the ABL-PDE (Abductive Learning for PDE discovery) approach to tackle the aforementioned challenges under the ABductive Learning (ABL) framework [28, 29]. ABL is a powerful paradigm that integrates data-driven machine learning with knowledge-driven logical reasoning in a balanced mutual promotion loop while maintaining the expressive power of both. Building on this framework, we design two components for ABL-PDE: a neural network-based learning module for derivative estimation and a First-order Logic (FOL)-based reasoning module for candidate term generation, along with a consistency optimization process to bridge the two components. Our approach balances the expressiveness and compactness of the hypothesis space in PDE discovery and enables reciprocal enhancement between the calculator and the library.

Our main contributions are summarized as follows:

- We propose a novel PDE discovery approach based on the ABL framework, transforming the previous unidirectional discovery pipeline into a bidirectional enhancement loop.
- We introduce a FOL knowledge base capable of representing a wide range of PDEs, substantially reducing redundancy and facilitating the incorporation of task-specific knowledge.
- We design a consistency optimization process to bridge the two components, which significantly enhances the robustness of structure identification, the accuracy of coefficient estimation, and the stability against hyperparameter variations in PDE discovery.

We conduct extensive experiments on various PDE discovery tasks across different noise levels. The experimental results demonstrate the effectiveness of our approach.

## 2 Related Work

According to whether the equation structure and its coefficients are determined synchronously or not, PDE discovery methods can be roughly categorized into two classes.

The first class comprises synchronous methods, most of which rely on sparse regression techniques. A pioneering approach is SINDy [5], which was extended by PDE-FIND [23] from ODEs to PDEs. Weak SINDy [11] further enhances noise robustness using weak formulation. Apart from explicit sparse regression, several synchronous methods such as PDE-Net [18, 19] and Bayesian hidden physics models [1] optimize derivatives and coefficients end-to-end. Although generally exhibiting superior performance in fitting physical fields, they struggle to distinguish noise from small-coefficient terms. D-CIPHER [16] extends the boundary of PDE discovery to learn parameters within nonlinear terms; this comes with a challenging optimization process, making it struggle to learn coupled PDE systems. Generally speaking, synchronous methods rely on predefined, typically polynomial libraries that can ensure term uniqueness, though such a compact library may lack expressiveness when encountering complex structures. Actually, they may even find it challenging to identify fractions.

The second class comprises asynchronous methods, often based on evolutionary algorithms, that do not need to be constrained by predefined libraries [8, 10, 12, 15, 25]. These methods first determine the equation's form by discrete optimization and then estimate the equation's coefficients. Although

such approaches incur higher computational costs due to discrete optimization, they mitigate the detrimental impact of interference terms on coefficient estimation while facilitating the incorporation of more flexible equation representation methods, such as binary trees [8, 15, 25]. Note there are great progresses in establishing theoretical foundation for evolutionary algorithms [30], such that many of these methods are not purely heuristics any more. These methods, however, often generate an excessive number of redundant and invalid PDEs within the hypothesis space. By contrast, through introducing a FOL knowledge base, our method achieves a better balance between expressiveness and compactness in PDE representation, while also facilitating the incorporation of prior knowledge.

## 3 Preliminaries

In this section, we present the problem setting and introduce the abductive learning framework.

### 3.1 Problem Setting

In this paper, we consider general dynamical systems governed by PDEs of the following form:

$$\partial_t u = \left[ f^{(1)}(u, \boldsymbol{x}), f^{(2)}(u, \boldsymbol{x}), \ldots, f^{(i)}(u, \boldsymbol{x}), \ldots \right] \cdot \boldsymbol{\xi}, \tag{1}$$

where $u : [0, T] \times \Omega \to \mathbb{R}$ is the solution function with time horizon $T$ and bounded spatial domain $\Omega \subset \mathbb{R}^d$, $t$ and $\boldsymbol{x}$ denote the temporal and spatial coordinate, each $f^{(i)}$ is a function of $u$ and $\boldsymbol{x}$, and $\boldsymbol{\xi}$ is their linear combination coefficient which is always sparse in practice. The data available is a set of triplets $\{(t_i, \boldsymbol{x}_i, \hat{u}(t_i, \boldsymbol{x}_i))\}_{i=1}^m$ representing $m$ temporal-spatial coordinates, with measurements that may contain noise. Each $f^{(i)}(u, \boldsymbol{x})$ collectively forms the candidate term library that will be gradually expanded during the execution of our approach. We call $f^{(i)}(u, \boldsymbol{x})$ an *expression term*.

**Definition 1** (Expression Term). An *expression term* is recursively formed by the following rule:

$$
\begin{aligned}
\langle \texttt{expr} \rangle \quad &::= \quad \langle \texttt{idpv} \rangle \mid \langle \texttt{dpv} \rangle \mid \partial_{\langle \texttt{idpv} \rangle} \langle \texttt{expr} \rangle \mid \langle \texttt{expr} \rangle \, \langle \texttt{op} \rangle \, \langle \texttt{expr} \rangle \\
\langle \texttt{op} \rangle \quad &::= \quad + \mid - \mid \times \mid /
\end{aligned}
$$

Here, $\langle \cdot \rangle$ denotes a nonterminal symbol, ::= indicates definition, and | signifies alternation. The nonterminal symbols $\langle \texttt{dpv} \rangle$ and $\langle \texttt{idpv} \rangle$ represent dependent and independent variables, respectively. Combining Equation (1) with expression terms, it suffices to represent a wide range of fundamental PDEs, such as the Diffusion equation, the Schrödinger equation, the Navier-Stokes equation, etc.

### 3.2 Abductive Learning

Abductive Learning (ABL) [28, 29] is a powerful paradigm that integrates data-driven induction from machine learning with knowledge-driven deduction from logical reasoning. A key challenge in this integration is that the discrete deductive process is non-differentiable. ABL overcomes this by using abductive reasoning to propagate the discrepancy between the logical deductions and data labels back to the machine learning model, which then serves as the supervision signal for its training.

Abductive reasoning, also known as abduction, is a fundamental mode of logical inference that seeks the best explanation for observations. A common approach to implement this process is through Abductive Logic Programming (ALP) [17], where an abductive logic program is formally defined by a triplet $(P, A, IC)$. $P$ is the background knowledge encoded as a logic program, $A$ is a set of abducible predicates representing hypotheses, and $IC$ is a set of first-order formulae called integrity constraints. Abductive explanation for an observation $O$ is a set $\Delta$ of ground atoms on abducible predicates $A$, such that the following conditions are met (where $\models$ denotes logical entailment):

(1) $P \cup \Delta \models O, IC$
(2) $P \cup \Delta$ is consistent

## 4 Our Approach

In this section, we first present an overview of our approach, ABL-PDE (ABductive Learning for PDE discovery), and then detail its components and the proposed consistency optimization process.

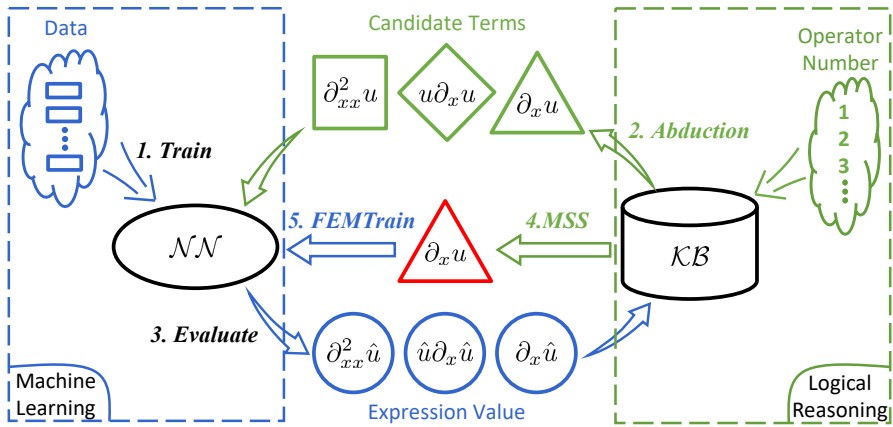

Figure 2: An illustration of ABL-PDE's framework.

## 4.1 Overview of ABL-PDE

As illustrated in Figure 2, ABL-PDE comprises two modules:

- **Machine Learning**. This module employs a neural network to fit the physical field and calculates derivatives through automatic differentiation.
- **Logical Reasoning**. This module incorporates an abductive logic program as the knowledge base and is responsible for generating candidate expression terms.

The discovery process commences with pretraining the neural network, during which no physical information is available (Step 1). Following this initial stage, an iterative deepening search is performed by progressively increasing the number of operators in the candidate terms. In each iteration, ABL-PDE first expands the candidate term library through abduction, utilizing the current number of operators as observations (Step 2). After this expansion, the abduced expression terms are transformed into numerical values (Step 3). Next, the proposed Monotone Subset Selection (MSS) algorithm is employed to identify terms with nonzero coefficients (Step 4). Subsequently, ABL-PDE estimates their coefficients and jointly optimizes the neural network through Finite Element Method-based Training (FEMTrain) (Step 5). Steps 4 and 5 together constitute the consistency optimization process, which is repeated until the discovered equation stabilizes and the coefficients converge.

## 4.2 Logic-based Knowledge Base

This part details our knowledge base KB, an abductive logic program $(P, A, IC)$.

**Background Knowledge** $P$. A straightforward way is to adopt the syntax rule in Definition 1 as the background knowledge $P$. Since $P \cup \Delta$ must be consistent, $P$ effectively defines the boundary of the search space by constraining $\Delta$ to be a valid expression term. This space, however, grows exponentially with the number of operators, posing a significant challenge. We observe that this vast space contains substantial redundancy, as many terms are mathematically equivalent despite differing in their syntactic representation. To mitigate this issue, we introduce *canonical form term*, which prune the search space by eliminating these equivalent representations.

**Definition 2** (Canonical Form Term). An expression term is called in *canonical form* if and only if it can be recursively generated by the following rules:

$$
\begin{aligned}
\langle \text{cfterm} \rangle &::= \langle \text{div} \rangle \mid \langle \text{mul} \rangle \mid \langle \text{add} \rangle \mid \langle \text{sub} \rangle \mid \langle \text{unit} \rangle \\
\langle \text{div} \rangle &::= \langle \text{mul} \rangle \,/\, \langle \text{add} \rangle \mid \langle \text{mul} \rangle \,/\, \langle \text{sub} \rangle \\
\langle \text{mul} \rangle &::= \langle \text{unit} \rangle \,\times\, \langle \text{mul} \rangle \mid \langle \text{unit} \rangle \\
\langle \text{add} \rangle &::= \langle \text{mul} \rangle \,+\, \langle \text{add} \rangle \mid \langle \text{mul} \rangle \\
\langle \text{sub} \rangle &::= \langle \text{add} \rangle \,-\, \langle \text{add} \rangle \\
\langle \text{unit} \rangle &::= \langle \text{dpv} \rangle \mid \langle \text{idpv} \rangle \mid \langle \text{par} \rangle \\
\langle \text{par} \rangle &::= \partial_{\langle \text{idpv} \rangle} \langle \text{dpv} \rangle \mid \partial_{\langle \text{idpv} \rangle} \langle \text{par} \rangle
\end{aligned}
$$

Table 1: General-purpose (C1, C2, C3) and domain-specific (C4, C5, C6) integrity constraints.

| | Integrity Constraint | Description |
|---|---|---|
| C1 | $\leftarrow \mathtt{add}(X, L) \wedge L = \mathtt{add}(Y, Z) \wedge X \succ Y.$ | Unique addition order |
| C2 | $\leftarrow \mathtt{mul}(X, L) \wedge L = \mathtt{mul}(Y, Z) \wedge X \succ Y.$ | Unique multiplication order |
| C3 | $\leftarrow \mathtt{sub}(X, Y) \wedge \mathtt{intersect}(X, Y).$ | Disjoint subtraction operands |
| C4 | $\leftarrow \mathtt{getPower}(\mathtt{mul}(X, Y), u, N) \wedge N > n.$ | Power limit |
| C5 | $\leftarrow \mathtt{getOrder}(\mathtt{par}(X, Y), x, M) \wedge M > m.$ | Derivative order limit |
| C6 | $\leftarrow \mathtt{getDependent}(\mathtt{par}(X, Y), Z) \wedge Z = v \wedge Y = y.$ | Continuity equation constraint |

The notation has the same meaning as in Definition 1. These rules impose structure constraints on expression terms derivable from each nonterminal symbol. We illustrate the constraint on partial derivatives ($\langle\mathtt{par}\rangle$) in Example 1; for detailed explanations of other rules, please refer to Appendix A.

**Example 1.** Given two dependent variables $u$ and $v$. The expression term $\partial_t(u \times v)$ is not in canonical form as it does not satisfy the rule of $\langle\mathtt{par}\rangle$ nor any other rules. However, $\partial_t(u \times v) = \partial_t u \times v + u \times \partial_t v$, which is the linear combination of canonical form terms $\partial_t u \times v$ and $u \times \partial_t v$. In Section 5, we prove that every expression term can be represented by a linear combination of canonical form terms.

In our implementation, $P$ is a logic program directly translated from the rules in Definition 2. Integrated with the operator number tracking capability in the deductive reasoning process, $P$ can leverage abductive reasoning to generate canonical form terms. [1]

**Abducible Predicates** $A$. In this work, the design of abducible predicates is not a focus. We simply introduce an auxiliary predicate to only allow abducing terms derivable from $\langle\mathtt{mul}\rangle$ and $\langle\mathtt{div}\rangle$ Since we consider linear combinations of expression terms, this does not compromise the expressiveness.

**Integrity Constraints** $IC$. Integrity constraints enable us to declaratively state various constraints on the structure of expression terms. Table 1 presents several examples of such constraints: the upper section lists general-purpose constraints, while the lower section includes domain-specific ones. The logical symbols are used as follows: '$\leftarrow$' denotes implication (with the left-hand side `False` omitted), '$\wedge$' represents conjunction, and '$\succ$' indicates symbolic order, which is a built-in predicate in most logic programming languages. C1 and C2 in Table 1 state that an addition or multiplication sequence is invalid if the former of two adjacent subterms is symbolically greater than the latter. C3 ensures that the two operands of a subtraction do not intersect. C4 and C5 respectively constrain the power of $u$ and the derivative order with respect to $x$ in a term. C6 is pertinent to the continuity equation. Specifically, in our experiment on the Navier-Stokes equation, knowing that 2D incompressible flow satisfies the continuity equation ($-\partial_x u = \partial_y v$) allows us to eliminate terms involving partial derivatives of $v$ with respect to $y$, since these terms can be systematically replaced by partial derivatives of $u$ with respect to $x$ (e.g., $v_y = -u_x$, $v_{yy} = -u_{xy}$). Please note that only the general-purpose constraints are included in KB to ensure a fair comparison with other methods.

### 4.3 Consistency Optimization

In this part, we first introduce the consistency measure and then present MSS and FEMTrain.

**Consistency Measure.** Given the operator number limit $L$ and the dataset $D$, the consistency measure $\mathrm{Con}_{L,D}(\mathtt{NN}_\theta, \mathtt{KB})$ of a neural network $\mathtt{NN}_\theta$ and a knowledge base KB is defined as follows:

$$-\left( \min_{\boldsymbol{\xi}, F} \mathrm{Res}(\mathtt{NN}_\theta, \boldsymbol{\xi}, F) + \lambda \cdot \mathrm{MSE}(\mathtt{NN}_\theta, D) \right)$$
$$\text{s.t.} \quad F \subset \bigcup_{O=1}^{L} \Delta_{\mathtt{KB},O}, \tag{2}$$

where $\theta$ denotes the neural network parameters, $\boldsymbol{\xi}$ is the coefficient vector as in Eq. (1), $F$ is a set of terms obtained from the abductive reasoning on KB for each operator number $O \leq L$, and $\lambda$ is a hyperparameter utilized to balance Res and MSE.

---

[1]Please refer to Appendix B for a pedagogical example illustrating the program and the abduction process.

Let $F = \{f^{(i)}(u, \boldsymbol{x})\}_{i=1}^{N_F}$, the physics-informed residual is defined as: [2]

$$\text{Res}(\text{NN}_\theta, \boldsymbol{\xi}, F) = \left\| \partial_t \text{NN}_\theta(t, \boldsymbol{x}) - \sum_{i=1}^{N_F} \xi_i \cdot f^{(i)}(\text{NN}_\theta(t, \boldsymbol{x}), \boldsymbol{x}) \right\|_2. \tag{3}$$

Let $D = \{(t_i, \boldsymbol{x}_i, \hat{u}(t_i, \boldsymbol{x}_i))\}_{i=1}^m$ as in Section 3.1, the mean square error is defined as:

$$\text{MSE}(\text{NN}_\theta, D) = \frac{1}{m} \sum_{i=1}^m \left( \text{NN}_\theta(t_i, \boldsymbol{x}_i) - \hat{u}(t_i, \boldsymbol{x}_i) \right)^2. \tag{4}$$

The minimal residual in Eq. (2) decreases monotonically as the size of $F$ increases. Thus, directly maximizing the consistency measure will lead to a trivial solution where $F = \bigcup \Delta_{\text{KB},O}$. Since the number of nonzero terms cannot be determined in advance, selecting an upper bound of $|F|$ is insufficient to balance sparsity and consistency. Thus we introduce *margin* and *k-level margin*.

**Definition 3** (Margin). Let $H$ be a set of expression terms, and $F \subseteq H$. The *margin* of $F$ with respect to $H$ is defined as:

$$\text{Margin}_H(F) = \frac{\|\text{Res}^*(\text{NN}_\theta, F)\|_2 - \|\text{Res}^*(\text{NN}_\theta, H)\|_2}{\|\text{Res}^*(\text{NN}_\theta, H)\|_2}, \tag{5}$$

where $\text{Res}^*(\text{NN}_\theta, \cdot)$ is the minimal residual in Eq. (2) with respect to $\boldsymbol{\xi}$.

**Definition 4** (k-Level Margin). Let $H$ be a set of expression terms, $k$ a positive integer, and $|\cdot|$ denote set cardinality. The *k-level margin* of $H$ is defined as

$$\text{LevelMargin}_H(k) = \min_{F' \subseteq H, |F'|=k} \text{Margin}_H(F'). \tag{6}$$

Intuitively, the margin of a subset $F$ quantifies how self-sufficient it is in approximating the dynamics $\partial_t \text{NN}_\theta$. A small margin suggests that the terms within $F$ are sufficient, while the terms outside it are negligible. Accordingly, the $k$-level margin measures the sufficiency of the best possible $k$-term model, indicating whether a parsimonious, $k$-term equation can accurately represent the system. As shown in Section 5, the level margin decreases monotonically as $k$ increases, which enables us to adaptively control the expression term number by applying a margin threshold.

**Monotone Subset Selection.** Under a given margin threshold, evaluating the consistency measure $\text{Con}_{L,D}(\text{NN}_\theta, \text{KB})$ with respect to the neural network parameters $\theta$ is time-consuming, as the complexity of the residual minimization subproblem in Eq. (2) is no less than that of the sparse regression under a cardinality constraint, which is generally NP-hard [20]. To reduce computational costs, we propose Monotone Subset Selection (MSS), a two-stage algorithm featuring an initial stage of rapid, large-scale coarse selection succeeded by a precise, small-scale refined search.

Algorithm 1 presents the details of MSS. In the first stage (cf. Line 2-3), MSS evaluates expression terms in the library and calls

---

**Algorithm 1** MSS

**Input:** Neural network `NN`, Term library `termLib`, Initial sparsity `initSp`, Margin threshold `magThd`
**Output:** Minimal residual set under `magThd`-margin `minSet`
 1: Initialization: `minSet` $\leftarrow \emptyset$
 2: `libVal` $\leftarrow$ Eval(`NN`, `termLib`)
 3: `subLib` $\leftarrow$ POSS(`libVal`, `termLib`, `initSp`)
 4: `left` $\leftarrow 1$, `right` $\leftarrow |\text{subLib}|$
 5: **while** `left` $\leq$ `right` **do**
 6:    `mid` $\leftarrow \lfloor (\text{left} + \text{right})/2 \rfloor$
 7:    `tmpMinSet`, `levMag` $\leftarrow$ Enum(`subLib`, `mid`)
 8:    **if** `levMag` $<$ `magThd` **then**
 9:       `minSet` $\leftarrow$ `tmpMinSet`, `right` $\leftarrow$ `mid` $- 1$
10:    **else**
11:       `left` $\leftarrow$ `mid` $+ 1$

---

Pareto Optimization for Subset Selection (POSS) [21, 30] to find a reduced term library `subLib` whose size should be no greater than the initial sparsity. POSS is a non-deterministic subset selection algorithm designed to minimize the residual in Eq. (2) under a cardinality constraint. In expectation, it is guaranteed to find a reasonably good solution within a polynomial number of queries to the objective function. This property makes it well-suited for the first stage, which focuses on rapidly identifying relevant terms rather than performing costly residual minimization. More details about

---

[2]Since the underlying PDE is expected to hold at every point within the domain, we are free to select the set of points for calculating the residual, which has been omitted for brevity.

POSS are provided in Appendix C. In the second stage (cf. Line 4-11), MSS employs binary search to find the smallest subset size for which the level margin is below the given threshold. Each time, it enumerates the subset of size `mid` to identify the one with the minimal residual and calculates the level margin, which is tractable within the reduced library size.

**Finite Element Method-based Training.** Once the set of active expression terms $F$ in Eq. (2) is identified via MSS for a given neural network parameters $\theta$, the next step of consistency optimization is to jointly optimize the network parameters $\theta$ and coefficient vector $\boldsymbol{\xi}$ to maximize the consistency measure. A standard approach for this sub-problem is to adopt the Physics-Informed Neural Networks (PINN) paradigm [22]. However, the efficacy of the PINN approach is often hindered by its sensitivity to hyperparameters, such as the loss weighting term $\lambda$ in Eq. (2) [13]. To enhance the stability of model training and coefficient estimation, we move beyond pointwise collocation and instead enforce the physics-informed residual in a weighted-average sense across the domain. This is achieved through a *Galerkin projection* [24] of the residual from Eq. (3) onto the space spanned by Finite Element Method (FEM) basis functions, leading to the following objective function:

$$\|\boldsymbol{M}(\partial_t \texttt{NN}_\theta(t, \boldsymbol{x}) - \sum_{i=1}^{N_F} \xi_i \cdot f^{(i)}(\texttt{NN}_\theta(t, \boldsymbol{x}), \boldsymbol{x}))\|_2, \tag{7}$$

where $\boldsymbol{M}$ is the mass matrix constructed from the inner product of FEM basis functions. Please refer to Appendix D for detailed definitions and derivations. The overall consistency optimization process alternates between MSS and FEMTrain until the set $F$ stabilizes and the coefficients $\boldsymbol{\xi}$ converge.

# 5 Theoretical Analysis

In this section, we first demonstrate that our logic-based knowledge base, KB, retains full expressive power, and then establish the soundness of MSS. Proofs are available in Appendix E.

We denote the abduction result of KB for a given operator number $i$ as $H_i$. Thus, the set of all possible abduction results is given by $H = \bigcup_{i=0}^{\infty} H_i$. Besides, we refer to the set of all expression terms as $S$.

**Definition 5** (Subsume)**.** A finite set of expression terms $F$ is said to *subsume* an expression term $s$ if there exists an expression term $s'$ such that:

- $s'$ can be derived from $s$ through a finite sequence of algebraic operations and differentiation.
- $s'$ can be symbolically expressed as a linear combination of expression terms in $F$, i.e., $s' = \sum_{f \in F} \alpha_f \cdot f$, where $\alpha_f$ are scalars.

As shown in Example 1, $\{\partial_x u \times v, u \times \partial_x v\}$ subsumes $\partial_x(u \times v)$.

**Theorem 1.** *For every $s \in S$, there exists a finite subset $H' \subset H$ such that $H'$ subsumes $s$.*

According to the PDE formulation in Eq. (1), it suffices to consider the expressiveness of a candidate term library up to a linear combination. Theorem 1 states that every expression term in $S$ has a mathematically equivalent form that is a linear combination of elements from $H$, which directly implies that the expressive power of KB is equivalent to that of the full expression term space $S$.

The soundness of MSS depends on the criterion utilized in its binary search, whose validity is ensured by the following monotonicity theorem.

**Theorem 2.** *For an expression term set $F$ and integers $1 \leq k_1 < k_2 \leq |F|$, it holds that:*

$$\text{LevelMargin}_F(k_1) \geq \text{LevelMargin}_F(k_2).$$

# 6 Experiments

In this section, we present empirical studies to answer the following questions:

1. How does ABL-PDE perform against contenders in PDE discovery?
2. How robust is ABL-PDE against hyperparameter selection?
3. How does each component of ABL-PDE contribute to its performance?

Table 2: PDE discovery comparison results. The evaluation metric is the sum of L2RE (%) on all coefficients. Number with a dagger (†) indicates the discovered equation has redundant terms. Results of ABL-PDE (w/o FEM) and ABL-PDE are the mean and standard deviation of 5 experiments with different $\lambda$ that used to balance the two sub-metrics in Eq. (2). We **bold** the best results in each task.

| Method | Burgers' Equation | | | Schrödinger Equation | | | Naiver-Stokes Equation | | |
|---|---|---|---|---|---|---|---|---|---|
| | 0 | 5% | 10% | 0 | 5% | 10% | 0 | 5% | 10% |
| PDERidge | 22.68 | 23.86 | 24.54 | 10.34 | 10.26 | 9.99 | **17.13** | 16.58 | 40.92 |
| DL-PDE++ | 22.68 | 23.86 | 24.54 | 10.34 | 10.26 | 9.99 | $57.25^\dagger$ | $37.96^\dagger$ | $98.52^\dagger$ |
| ABL-PDE (w/o CO) | 22.68 | 23.86 | 24.54 | $42.16^\dagger$ | $44.67^\dagger$ | $48.76^\dagger$ | **17.13** | 17.99 | 40.92 |
| ABL-PDE (w/o FEM) | $13.69 \pm 2.81$ | $14.65 \pm 2.67$ | $15.04 \pm 3.87$ | $3.05 \pm 0.94$ | $3.27 \pm 1.09$ | $2.87 \pm 1.07$ | $30.88 \pm 5.24$ | $26.47 \pm 5.00$ | $34.28 \pm 7.50$ |
| ABL-PDE | $\mathbf{2.18 \pm 1.06}$ | $\mathbf{2.79 \pm 0.13}$ | $\mathbf{4.12 \pm 0.34}$ | $\mathbf{1.03 \pm 0.10}$ | $\mathbf{1.12 \pm 0.06}$ | $\mathbf{1.08 \pm 0.29}$ | $18.02 \pm 0.08$ | $\mathbf{15.22 \pm 0.08}$ | $21.27 \pm 1.72$ |

## 6.1 Experimental Setup

We conduct experiments on three types of discovery tasks: (i) one-dimensional PDE: Burgers' Equation, (ii) one-dimensional PDE system: Schrödinger Equation, (iii) two-dimensional PDE system: Navier-Stokes Equation. Tasks (ii) and (iii) each involve two coupled equations. We use data from public datasets [2], incorporating 5% and 10% Gaussian noise to simulate inaccurate measurements as in [27]. We use the $L_2$ Relative Error (L2RE, %) as in [11] to measure the quality of coefficient estimation and physical field approximation. More details are specified in Appendix F. Code is available at: `https://github.com/AbductiveLearning/ABL-PDE`.

## 6.2 Compared Methods

We compare ABL-PDE with four methods: 1) **PDERidge**: We build this strong baseline by assuming a known PDE structure and applying ridge regression to estimate coefficients. This represents the best possible performance for most methods that focus on enhancing structure identification [8, 10, 16]. 2) **DL-PDE++**: DL-PDE [27] is a representative method among synchronous approaches that discussed in Section 2. We enhance it by applying a normalization trick and tuning the sparsity threshold to retain all correct terms while minimizing redundancy. 3) **ABL-PDE (w/o Con)**: It utilizes the pre-trained neural network and MSS algorithm to discover PDE, without the subsequent consistency optimization. 4) **ABL-PDE (w/o FEM)**: It employs the residual in Eq. (3) instead of the FEM residual in Eq. (7) during the consistency optimization. To ensure a fair comparison, we use the same pre-trained neural network as in ABL-PDE to compute derivatives for the first two methods.

To verify the effectiveness of our logic-based knowledge base KB in reducing redundancy, we compare it with two baselines: 1) **Full Space**: The full space of expression terms as defined in Definition 1. 2) **Canonical Form**: It contains all expression terms in the canonical form, which lacks abducible predicates and integrity constraints compared to KB.

## 6.3 PDE Discovery Comparison

In this part, we seek to answer the first two questions. Table 2 presents the results of structure identification and coefficient estimation. Number with a dagger (†) indicates the discovered equation has redundant terms (See Appendix G.1 for the specific form). The performance of DL-PDE++ and ABL-PDE (w/o CO) is equivalent to PDERidge when the discovered structure matches the ground truth, due to their shared neural network for derivative computation. As shown in the table, ABL-PDE correctly identified the PDE structure across all three tasks under varying noise levels, achieving the best coefficient estimation in 8 out of 9 cases and ranking second only to the structure-known baseline, PDERidge, in the remaining one case. Notably, this performance was achieved utilizing the *same* initial sparsity of 10 and the *same* margin threshold of 0.05 for all three tasks. ABL-PDE has only one remaining hyperparameter, which will be discussed in Section 6.4. In our experiments, despite the simple structure of Burgers' equation, it remains challenging for coefficient estimation. The inferior performance of compared methods is primarily attributed to the formation of sharp shock waves in the solution, which can only be captured by an extremely dense mesh that is impractical in most real-world applications. ABL-PDE effectively mitigates this issue via consistency optimization, which utilizes the discovered physical information, even if it is initially incorrect, to improve the derivative estimation, leading to a 5x to 10x reduction in L2RE.

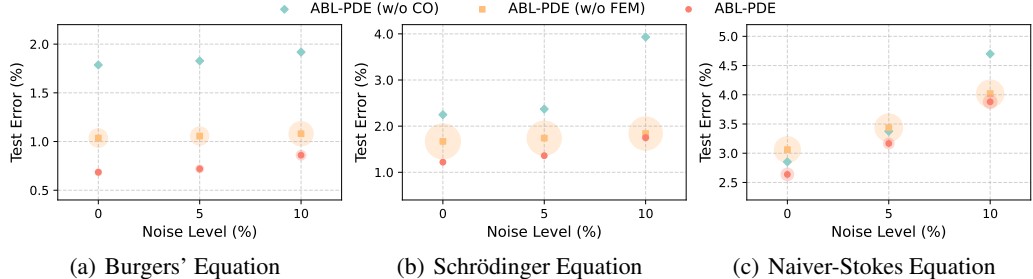

(a) Burgers' Equation      (b) Schrödinger Equation      (c) Naiver-Stokes Equation

Figure 3: Test error comparison results. The evaluation metric is the sum of L2RE on each component of the predicted physical field. Results of ABL-PDE (w/o FEM) and ABL-PDE show the mean and standard deviation of 5 experiments under different $\lambda$ used in Eq. (2). The area of the shadowed circle represents the magnitude of the standard deviation. Detailed results are provided in Appendix G.2.

Beyond its robustness to noise, ABL-PDE is also resilient to hyperparameter selection. As shown in Table 2, the discovered equation of ABL-PDE (w/o CO) contains redundant terms in several cases. This redundancy stems from both the neural network's imperfect approximation to the physical field and, more critically, from a misalignment between the margin threshold and the data. Nevertheless, ABL-PDE successfully eliminates these terms through consistency optimization, demonstrating its notable robustness to sub-optimal hyperparameter choices. A by-product of our method is the improvement in the neural network's generalizability. Figure 3 presents the L2RE of model prediction on the noise-free test data. Compared with the best performance achievable by supervised training alone (cf. ABL-PDE (w/o CO)), ABL-PDE consistently outperforms it across all tasks. Initially, the neural network performs suboptimally, and the candidate term library is too expansive. Consistency optimization progressively clarifies the ambiguity within the library, leading to a refined equation that, in turn, provides physical insights to improve neural network.

## 6.4 Ablation Study

**Influence of FEMTrain.** We investigate the effect of FEMTrain by comparing the performance of our method with and without the utilization of FEM residual. Selecting a hyperparameter, e.g., $\lambda$ in Eq. (2), to balance the physical information and the supervision signal has long been an existing issue with physics-informed machine learning [13]. Therefore, we conduct five experiments for each task with hyperparameters: $[0.5\lambda, 0.75\lambda, \lambda, 1.25\lambda, 1.5\lambda]$. The mean and standard deviation of coefficient errors are presented in Table 2, and the results of test errors are visualized in Figure 3. In addition to the superior performance of ABL-PDE in coefficient estimation and test error, its stability to hyperparameter variations is particularly noteworthy. In Figure 3, we use the area of the shadowed circle to reflect the magnitude of the standard deviation. Compared to ABL-PDE (w/o FEM), the standard deviation of ABL-PDE is reduced by approximately one order of magnitude. This enhanced stability is crucial for practical scientific discovery, as finding an effective hyperparameter is much easier than finding the optimal one.

**Influence of KB.** We use the one-dimensional PDE with a single dependent variable as a showcase to demonstrate the effectiveness of KB, i.e., the abductive logic program based on the canonical form, in constraining the hypothesis space. As the full space of expression terms is enormous, we use dynamic programming to compute its size, and the sizes of Canonical Form and ABL-PDE are counted by the implemented logic programs, respectively. Figure 4 illustrates the logarithmic growth curve of the candidate term number as the operator number limit increases. Compared to the full space of expression terms, the proposed canonical form reduces the problem size to a computationally

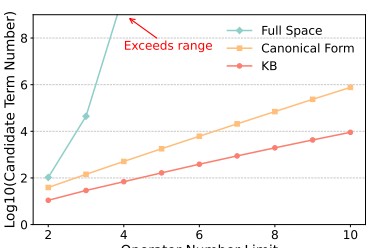

Figure 4: Logarithmic growth curve of the candidate term number as the operator number limit increases.

feasible scale. Nevertheless, KB still demonstrates a reduction by a constant factor in the exponent, making it tractable for POSS. As terms with 10 operators can construct highly complex equations, our method can be applied to most scenarios.

## 7    Conclusion

In this paper, we propose a novel PDE discovery method based on the abductive learning framework, aiming at leveraging the logic-based knowledge base and the neural network in a reciprocal way. Specifically, we introduce the ABL-PDE (ABductive Learning for PDE discovery) approach, which alternatively refines the knowledge base by utilizing a neural network to access data and enhances the neural network's performance by integrating the physical information provided by the knowledge base. Experimental results demonstrate the robustness of our approach in PDE structure identification, the accuracy in coefficient estimation, and the stability against hyperparameter variations. Furthermore, ABL-PDE is a general-purpose approach with sufficient flexibility in implementation, e.g., the neural network can adopt any advanced architecture, and the knowledge base can incorporate domain-specific information by declaring new integrity constraints. The limitation of ABL-PDE lies in its restriction to learning linear combination coefficients, which prevents it from handling unknown parameters within nonlinear terms. In future work, we will try to construct a parameterized candidate term library and incorporate advanced techniques for joint parameter optimization.

## 8    Acknowledgements

This research was supported by the Noncommunicable Chronic Diseases-National Science and Technology Major Project (2024ZD0531802), the National Natural Science Foundation of China (62202218), and the Jiangsu Science Foundation Leading-edge Technology Program (BK20232003).

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

## A  Detailed Explanation of the Canonical Form

The canonical form of expression terms is defined by the BNF grammar presented in Definition 2. This grammar uses standard notation: $\langle \cdot \rangle$ for nonterminal symbols, $::=$ for production rules (indicating that the nonterminal on the left can be replaced by the expression on the right), and $|$ signifies alternation between possible forms. The notation used here is consistent with that in Definition 1. In this grammar, nonterminal symbols define the valid structure for terms corresponding to specific operators or types: $\langle \texttt{cfterm} \rangle$ for any canonical form term, $\langle \texttt{div} \rangle$ for division ($/$), $\langle \texttt{mul} \rangle$ for multiplication ($\times$), $\langle \texttt{add} \rangle$ for addition ($+$), $\langle \texttt{sub} \rangle$ for subtraction ($-$), $\langle \texttt{par} \rangle$ for partial differentiation ($\partial$), $\langle \texttt{unit} \rangle$ for base multiplicative/derivative units, and $\langle \texttt{var} \rangle$, $\langle \texttt{dpv} \rangle$, $\langle \texttt{idpv} \rangle$ represent variables.

The production rules in Definition 2 define the set of expression terms that are in canonical form by imposing syntactic constraints on their structure. We explain the meaning of these rules as follows:

- The top-level rule, $\langle \texttt{cfterm} \rangle ::= \langle \texttt{div} \rangle \mid \langle \texttt{mul} \rangle \mid \langle \texttt{add} \rangle \mid \langle \texttt{sub} \rangle \mid \langle \texttt{unit} \rangle$, specifies that any expression in canonical form must be derivable from one of these five main nonterminals or be a base unit.

- The rule for division, $\langle \texttt{div} \rangle ::= \langle \texttt{mul} \rangle / \langle \texttt{add} \rangle \mid \langle \texttt{mul} \rangle / \langle \texttt{sub} \rangle$, defines the valid structure for division terms ($/$). It requires the numerator to be a term derivable from $\langle \texttt{mul} \rangle$ and the denominator to be a term derivable from either $\langle \texttt{add} \rangle$ or $\langle \texttt{sub} \rangle$. For example, a term corresponding to $(u + v)/(u - v)$ would be considered invalid under this rule because its numerator, $(u + v)$, is derivable from $\langle \texttt{add} \rangle$, not $\langle \texttt{mul} \rangle$.

- The recursive rule for multiplication, $\langle \texttt{mul} \rangle ::= \langle \texttt{unit} \rangle \times \langle \texttt{mul} \rangle \mid \langle \texttt{unit} \rangle$, defines the structure of multiplication terms ($\times$). The base case is any term derivable from $\langle \texttt{unit} \rangle$. The recursive step requires a term derivable from $\langle \texttt{unit} \rangle$ on the left of "$\times$" and a term derivable from $\langle \texttt{mul} \rangle$ on the right. This imposes a specific structure on multiplication chains; for instance, a product of three terms like $u \times v \times w$ can only be derived in the form $\langle \texttt{unit} \rangle \times \langle \texttt{mul} \rangle$, where the second term derivable from $\langle \texttt{mul} \rangle$ is $v \times w$. This enforces the structure $u \times (v \times w)$, preventing syntactically distinct but equivalent forms like $(u \times v) \times w$.

- The recursive rule for addition, $\langle \texttt{add} \rangle ::= \langle \texttt{mul} \rangle + \langle \texttt{add} \rangle \mid \langle \texttt{mul} \rangle$, similarly defines the structure of addition terms ($+$). It allows a base case of a term derivable from $\langle \texttt{mul} \rangle$, and a recursive step $\langle \texttt{mul} \rangle + \langle \texttt{add} \rangle$. This ensures a specific structure for addition chains; for example, an expression like $t + u \times v + w$ (where $u \times v$ is derivable from $\langle \texttt{mul} \rangle$) can only be derived as $\langle \texttt{mul} \rangle + \langle \texttt{add} \rangle$, where the first term derivable from $\langle \texttt{mul} \rangle$ is $t$ and the term derivable from $\langle \texttt{add} \rangle$ is $u \times v + w$. This enforces the structure $t + (u \times v + w)$, preventing forms like $(t + u \times v) + w$.

- The rule for subtraction, $\langle \texttt{sub} \rangle ::= \langle \texttt{add} \rangle - \langle \texttt{add} \rangle$, defines subtraction terms ($-$), requiring both the left and right operands to be derivable from $\langle \texttt{add} \rangle$. This enforces a specific structure; for example, $u + v - w$ (where $u + v$ and $w$ are derivable from $\langle \texttt{add} \rangle$) can only be derived as $(u + v) - w$, not $u - (w - v)$.

- The rule $\langle \texttt{unit} \rangle ::= \langle \texttt{dpv} \rangle \mid \langle \texttt{idpv} \rangle \mid \langle \texttt{par} \rangle$ defines terms derivable from $\langle \texttt{unit} \rangle$. These are referred to as unit expressions and serve as the base elements for multiplication and differentiation arguments.

- The rule for partial differentiation, $\langle \texttt{par} \rangle ::= \partial_{\langle \texttt{idpv} \rangle} \langle \texttt{dpv} \rangle \mid \partial_{\langle \texttt{idpv} \rangle} \langle \texttt{par} \rangle$, defines partial derivative terms ($\partial$). It requires the derivative operator to be applied to a term derivable from either $\langle \texttt{dpv} \rangle$ or $\langle \texttt{par} \rangle$, and differentiation is always with respect to a term derivable from $\langle \texttt{idpv} \rangle$ (an independent variable). This imposes strong constraints, for instance, forbidding derivatives of products or sums as single terms, such as $\partial_x (u \times v)$, which would be invalid because $u \times v$ is neither derivable from $\langle \texttt{dpv} \rangle$ nor $\langle \texttt{par} \rangle$.

## B  A Pedagogical Example of Logic-based Knowledge Base

We provide a pedagogical example to elucidate the logic-based knowledge base and the abduction process discussed in the main text. As a pedagogical example, this knowledge base considers three basic operators ($+, -, /$), and the full implementation (`reasoning/expr.pl`) is provided in the code of the supplementary material. By querying `?- write_to_file.` in SWI-Prolog, we can generate all expression terms with up to a given number of operators (2 in this code). Abduced expressions are

all in the form similar to `mul(u,mul(v,v))`. A single integrity constraint is employed to ensure that multiplication expressions are in a canonical order, illustrating how various human knowledge can be declaratively stated through such constraints. The declarative nature simplifies the task by allowing the user to specify desired properties directly, with the generation of constraint-satisfying expressions entirely managed by the Prolog prover. The implementation of this simplified knowledge base and abduction process is detailed in the following Prolog code.

```prolog
%%%%%%%%%%%%%%%%%%%%%%
% Background Knowledge
%%%%%%%%%%%%%%%%%%%%%%

% Declare the dependent variables
dep_var([u, v]).

% Rule for fraction expressions: (X/Y)
% Cur_num: Counter for the number of operations used so far
% Op_limit: Maximum operations allowed
fraction_expr(frac(X, Y), Cur_num, Op_limit) -->
    ['('],
    multiplication_expr(X, Cur_num1, Op_limit - 1), % Parse
        numerator (X)
    ['/'],
    addition_expr(Y, Cur_num2, Op_limit - Cur_num1 - 1), %
        Parse denominator (Y)
    {Cur_num is Cur_num1 + Cur_num2 + 1}, % Calculate total
        operations used (adding 1 for fraction)
    [')'].

% Rule for multiplication expressions: (X*Y)
% First clause handles actual multiplication
multiplication_expr(mul(X, Y), Cur_num, Op_limit) -->
    ['('],
    unit_expr(X, Cur_num1, Op_limit - 1), % Parse first operand
        (X)
    [*],
    multiplication_expr(Y, Cur_num2, Op_limit - Cur_num1 - 1),
        % Parse second operand (Y)
    {Cur_num is Cur_num1 + Cur_num2 + 1},
    {\+constraint(mul(X, Y))}, % Check that this multiplication
        doesn't violate integrity constraints
    [')'].

% Second clause for multiplication_expr
% This is how the recursion terminates
multiplication_expr(X, Cur_num, Op_limit) -->
    {0 =< Op_limit},
    unit_expr(X, Cur_num, Op_limit).

% Rule for addition expressions: (X+Y)
addition_expr(add(X, Y), Cur_num, Op_limit) -->
    ['('],
    multiplication_expr(X, Cur_num1, Op_limit - 1), % Parse
        first operand (X)
    [+],
    addition_expr(Y, Cur_num2, Op_limit - Cur_num1 - 1), %
        Parse second operand (Y)
    [')'],
    {Cur_num is Cur_num1 + Cur_num2 + 1}.

% Second clause for addition_expr
% This is how the recursion terminates
addition_expr(X, Cur_num, Op_limit) -->
```

```prolog
    multiplication_expr(X, Cur_num, Op_limit).

% Rule for basic units
% This is a terminal rule that matches a single variable from
    the dependent variable list
unit_expr(X, Cur_num, Op_limit) -->
    {
        Cur_num is 0,
        Cur_num =< Op_limit,
        dep_var(DepVarList), member(X, DepVarList) % X must be
            one of the dependent variables
    },
    [X].

%%%%%%%%%%%%%%%%%%%%%%%%%
% Abducible Predicates
%%%%%%%%%%%%%%%%%%%%%%%%%
% Decide which terms can be utilized to explain the
    observation, i.e., which values of X can satisfy
    expression_term(X, Op_limit)
expression_term(X, Op_limit) --> fraction_expr(X, _, Op_limit).
expression_term(X, Op_limit) --> multiplication_expr(X, _,
    Op_limit).

%%%%%%%%%%%%%%%%%%%%%%%%%
% Integrity Constraints
%%%%%%%%%%%%%%%%%%%%%%%%%
% Constraint to enforce a canonical ordering of multiplication
    terms
% This prevents generating both (a*b) and (b*a) which are
    mathematically equivalent
constraint(mul(X, Y)) :- extract_first_mul_term(Y, FirstTerm),
    X @> FirstTerm.

% Helper predicate to extract the first term in a
    multiplication chain
% For mul(a, mul(b, c)), it returns 'a'
% For a simple term like 'u', it returns 'u'
extract_first_mul_term(Term, FirstTerm) :-
    (   functor(Term, mul, 2) % Check if Term is a
        multiplication
    ->  arg(1, Term, FirstTerm) % If yes, extract first argument
    ;   FirstTerm = Term % If not, the term itself is the first
        term
    ).

% Utility predicate to write all generated expressions to a file
write_to_file :-
    open('out.txt', write, Stream),
    findall([X, Y], phrase(expression_term(X, 2), Y), L), %
        Generate all expressions with max 2 operators
    forall(member([X, Y], L),
        (write(Stream, X), nl(Stream),
            write(Stream, Y), nl(Stream))),
    close(Stream).
```

## C   A Brief Introduction to POSS

POSS leverages Pareto Optimization to solve the following subset selection problem:

**Algorithm 2** POSS

---

**Input:** all variables $V = \{X_1, \ldots, X_n\}$, a given criterion $f$ and an integer parameter $k \in [1, n]$
**Parameter:** the number of iterations $T$ and an isolation function $I : \{0, 1\}^n \to \mathbb{R}$
**Output:** a subset of $V$ with at most $k$ variables
 1: Let $s = \{0\}^n$ and $P = \{s\}$.
 2: Let $t = 0$.
 3: **while** $t < T$ **do**
 4:     Select $s$ from $P$ uniformly at random.
 5:     Generate $s'$ from $s$ by flipping each bit of $s$ with probability $1/n$.
 6:     **if** $\nexists z \in P$ such that $I(z) = I(s')$ and

$$(z.o_1 < s'.o_1 \wedge z.o_2 \leq s'.o_2) \text{ or } (z.o_1 \leq s'.o_1 \wedge z.o_2 < s'.o_2)$$

    **then**
 7:         $Q = \{z \in P \mid I(z) = I(s') \wedge s'.o_1 \leq z.o_1 \wedge s'.o_2 \leq z.o_2\}$.
 8:         $P = (P \setminus Q) \cup \{s'\}$.
 9:     $t = t + 1$.
10: **Return** $\arg\min_{s \in P, |s| \leq k} f(s)$

---

**Definition 6** (Sparse Regression). Given all observation variables $V = \{X_1, \ldots, X_n\}$, a predictor variable $Z$ and a positive integer $k$, define the mean squared error of a subset $F \subseteq V$ as

$$MSE_{Z,F} = \min_{\alpha \in \mathbb{R}^{|F|}} \mathbb{E}\left[\left(Z - \sum_{i \in F} \alpha_i X_i\right)^2\right].$$

Sparse regression is to find a set of at most $k$ variables minimizing the mean squared error, i.e.,

$$\arg\min_{F \subseteq V} MSE_{Z,F} \quad \text{s.t.} \quad |F| \leq k.$$

We use it to roughly select the subset that contains possible relevant terms. The following is the pseudo-code of POSS:

In Algorithm 2, $o_1$ represents residual and $o_2$ represents the subset size. $I : \{0, 1\}^n \to \mathbb{R}$ determines if two solutions are allowed to be compared: they are comparable only if they have the same isolation function value.

## D   More Details on FEMTrain

### D.1   A Brief Introduction to FEM

Finite Element Method (FEM) is a well-established numerical method that provides a systematic framework for approximating solutions to PDEs.

We use a scalar field $u$ with one spatial dimension, $[0, L]$, as a showcase to explain the process of using FEM to solve the following type PDE:

$$\partial_t u = f\left(u, x, \partial_x u, \partial_x^2 u, \ldots\right). \tag{8}$$

The first step to apply FEM is to discretize the spatial domain into smaller, finite subdomains called elements. Given an ordered set of points $X = \{x_1, x_2, \ldots, x_n\}$ with $x_1 = 0$ and $x_n = L$, they split the domain into $n - 1$ elements: $[x_1, x_2], \ldots, [x_{n-1}, x_n]$. The second step is to construct a set of basis functions $\{\phi_1, \phi_2, \ldots, \phi_n\}$ to approximate the solution $u$. $\phi_i$ is defined as follows:

$$\phi_i(x) = \begin{cases} 0, & \text{if } x < x_{i-1} \text{ or } x > x_{i+1}, \\[2mm] \frac{x - x_{i-1}}{x_i - x_{i-1}}, & \text{if } x_{i-1} \leq x < x_i, \\[2mm] \frac{x_{i+1} - x}{x_{i+1} - x_i}, & \text{if } x_i \leq x \leq x_{i+1}. \end{cases}$$

The solution $u$ is then approximated by $\tilde{u}$ defined as:

$$\tilde{u}(t, x) = \sum_{i=1}^{n} \alpha_i(t)\phi_i(x), \tag{9}$$

where $\alpha_i(t) = u(t, x_i)$. The third step is to transform Eq. (8) into *weak formulation* by multiplying both sides with each basis function and integrating over the domain:

$$\int_0^L \partial_t u \cdot \phi_i \, \mathrm{d}x = \int_0^L f(u, x, \ldots) \cdot \phi_i \, \mathrm{d}x. \tag{10}$$

Substitute Eq. (9) into Eq. (10), we get a linear system:

$$\boldsymbol{M}\frac{\mathrm{d}}{\mathrm{dt}}\boldsymbol{\alpha}(t) = \boldsymbol{F}(t), \tag{11}$$

where $\boldsymbol{M}_{ij} = \int_0^L \phi_i \cdot \phi_j \, \mathrm{d}x$ is the so-called mass matrix in finite element analysis, $\boldsymbol{\alpha}(t)$ is the vector of coefficients in Eq. (9), and $F_i(t) = \int_0^L f(\tilde{u}, x, \ldots) \cdot \phi_i \, \mathrm{d}x$. Eq. (11) is an Ordinary Differential Equation (ODE) system of $\boldsymbol{\alpha}$. Therefore, given the initial state, we can calculate $\boldsymbol{F}$ and forward $\boldsymbol{\alpha}$ iteratively, thus forecasting $\tilde{u}$ at any time.

### D.2  FEM-based Residual

Firstly, we discretize the spatial domain $\Omega$ into a set of cubic elements $\mathcal{T} = \{\Omega_i \subset \mathbb{R}^d\}_{i=1}^{N_\Omega}$. Let $X = \{\boldsymbol{x}_i \in \mathbb{R}^d\}_{i=1}^N$ denote their vertices, and $\Phi = \{\phi_i : \Omega \to \mathbb{R}\}_{i=1}^N$ for the corresponding basis functions. Secondly, we use $\Phi$ to approximate $\partial_t u(t, \boldsymbol{x}) \triangleq f^{(0)}(u(t, \boldsymbol{x}), \boldsymbol{x})$ and expression terms $\{f^{(i)}(u(t, \boldsymbol{x}), \boldsymbol{x})\}_{i=1}^{N_F}$ identified by MSS as follows:

$$\tilde{f}^{(i)}(u(t, \boldsymbol{x}), \boldsymbol{x}) = \sum_{j=1}^{N} \alpha_j^{(i)}(t)\phi_j(\boldsymbol{x}), i = 0, 1, \ldots, N_F, \tag{12}$$

where $\alpha_j^{(i)}(t) = f^{(i)}(u(t, \boldsymbol{x}_j), \boldsymbol{x}_j)$, which can be evaluated through the pre-trained neural network. Thirdly, we transform $f^{(0)} = \sum_{i=1}^{N_F} \xi_i \cdot f^{(i)}$ into the weak formulation:

$$\langle f^{(0)}, \phi_j \rangle = \sum_{i=1}^{N_F} \xi_i \cdot \langle f^{(i)}, \phi_j \rangle, j = 1, 2, \ldots, N, \tag{13}$$

where $\xi_i$ is the coefficient as introduced in Eq. (1), and $\langle u, v \rangle = \int_\Omega u \cdot v \, \mathrm{d}\boldsymbol{x}$. Substituting (12) into (13) at a given time point $t$ (omitted for brevity) yields the linear system:

$$\boldsymbol{M}\boldsymbol{\alpha}^{(0)} = \boldsymbol{M}\boldsymbol{A}\boldsymbol{\xi}, \tag{14}$$

where $\boldsymbol{M}_{ij} = \langle \phi_i, \phi_j \rangle$, $\boldsymbol{\alpha}^{(i)} = (\alpha_0^{(i)}, \alpha_1^{(i)}, \ldots, \alpha_N^{(i)})^T$, $\boldsymbol{A} = \begin{bmatrix} \boldsymbol{\alpha}^{(1)} \, \boldsymbol{\alpha}^{(2)} \, \ldots \, \boldsymbol{\alpha}^{(N_F)} \end{bmatrix}$, and $\boldsymbol{\xi} = (\xi_0, \xi_1, \ldots, \xi_{N_F})^T$. Since $\{\boldsymbol{\alpha}^{(i)}\}_{i=0}^{N_F}$ are outputs of the neural networks, the following FEM-based residual can be utilized to optimize neural network parameters $\theta$ and PDE coefficients $\boldsymbol{\xi}$:

$$\|\boldsymbol{M}(\boldsymbol{\alpha}^{(0)} - \boldsymbol{A}\boldsymbol{\xi})\|_2 \tag{15}$$

Replacing $\mathrm{Res}(\mathrm{NN}_\theta, \boldsymbol{\xi}, F)$ in Eq. (2) with this residual gives the loss function utilized in FEMTrain.

## E  Proofs

### E.1  Theorem 1

**Lemma 1.** *Let $e_1$ and $e_2$ be two expressions generated by* ``. *Then, $e_1 \times e_2$ can be rewritten as an expression generated by* ``.

*Proof.* According to Definition 2, $e_1$ can be expressed as $a - b$, $e_2$ can be expressed as $c - d$, where $a$, $b$, $c$, $d$ are expressions generated by `<add>`, `<mul>`, `<par>`, `<dep>`, or `<indep>`. Without loss of generality, we only need to consider the `<add>` case, as all other cases are simply degenerate forms of this one. Since $(a - b) \times (c - d) = (ac + bd) - (bc + ad)$, we only need to prove $ac + bd$ and $bc + ad$ can be rewritten as expressions generated by `<add>`. Since $a$ and $c$ are summations of multiplication or unit expressions, it's obvious that we can rewrite $ac$ as an expression generated by `<add>` according to basic algebraic laws, so as $bd$, $bc$, and $ad$. Therefore, $(ac + bd) - (bc + ad)$ is a subtraction between two expressions generated by `<add>`, which can be rewritten as an expression generated by ``.

$\square$

**Lemma 2.** *Let $e_1$ and $e_2$ be two expressions generated by `<div>`. Then, $e_1 \times e_2$ can be rewritten as an expression generated by `<div>`.*

*Proof.* Without loss of generality, we assume $e_1 = \frac{a}{b-c}$, $e_2 = \frac{d}{e-f}$, where $a$ and $d$ are expressions generated by `<mul>`; $b$, $c$, $e$, and $f$ are expressions generated by `<add>`. According to Lemma 1, the denominator of $e_1 \times e_2$ can be written as an expression generated by ``. The numerator of $e_1 \times e_2$ can be rewritten as an expression generated by `<mul>`, thus $e_1 \times e_2$ can be rewritten as an expression generated by `<div>`.

$\square$

**Lemma 3.** *Let $e_1$ and $e_2$ be two expressions generated by `<div>`. Then, the numerator and denominator of $e_1$ `op` $e_2$, where `op` $\in \{+, -, /\}$ can all be rewritten as expressions generated by ``.*

*Proof.* Without loss of generality, we assume $e_1 = \frac{a}{b-c}$, $e_2 = \frac{d}{e-f}$, where $a$ and $d$ are expressions generated by `<mul>`; $b$, $c$, $e$, and $f$ are expressions generated by `<add>`. We prove the "+" case and others are similar. Since $e_1 + e_2 = \frac{a(e-f)+d(b-c)}{(b-c)(e-f)}$, according to Lemma 1, $a(e-f) + d(b-c)$ and $(b-c)(e-f)$ are all expressions generated by ``.

$\square$

**Lemma 4.** *Let $e$ be an expression generated by `<div>`. Then, $\frac{\partial e}{\partial x}$ can be subsumed by a set $H' \subset H$, where $x$ is an arbitrary independent variable.*

*Proof.* Without loss of generality, we assume $e = \frac{a}{b-c}$, where $a$ is an expression generated by `<mul>`; $b$ and $c$ are expressions generated by `<add>`. According to differentiation rule, $\partial_x \frac{a}{b-c} = \frac{\partial_x a(b-c) - a \partial_x(b-c)}{(b-c)^2}$. According to Lemma 1, $(b-c)^2$ is an expression generated by ``. According to the differentiation rule and basic laws of multiplication, the numerator is an expression generated by ``. Thus, $\frac{\partial e}{\partial x}$ can be subsumed by a set $H' \subset H$.

$\square$

**Proof of Theorem 1.** We use structural induction to prove Theorem 1.

1. **Base Case:** For each expression term $s$ with zero operator, there exists a term $h \in H$ such that $s = h$.

   This case is obviously true.

2. **Inductive Hypothesis:** Assume that for each expression term $s$ with no greater than $n$ operators, there exists a subset of $H' \subset H$ such that $H'$ subsumes $s$.

3. **Inductive Step:** Show that for each expression term $s$ with $n + 1$ operators, there exists a subset of $H' \subset H$ such that $H'$ subsumes $s$.

   If $s$ is an addition or subtraction, according to the inductive hypothesis, the conclusion is obviously true.

   If $s$ is a multiplication, assume $s = a \times b$, according to the inductive hypothesis, $a$ and $b$ can be subsumed by two sets $H_a \subset H$ and $H_b \subset H$ respectively. Without loss of generality, we assume each element of $H_a$ and $H_b$ are all expressions generated by `<div>`. According to the distributive law of multiplication and Lemma 2, the conclusion holds.

If $s$ is a division, let $s = a/b$, according to the inductive hypothesis, $a$ and $b$ can be subsumed by two sets $H_a \subset H$ and $H_b \subset H$ respectively. Without loss of generality, we assume each element of $H_a$ and $H_b$ are all expressions generated by `<div>`. Then, we only need to prove $\frac{c+d}{e+f}$ can be subsumed by a set $H' \subset H$, where $c$, $d$, $e$, and $f$ are all expressions generated by `<div>`. Other cases are all degenerated cases. According to Lemma 3, $\frac{c+d}{e+f}$ can be rewrite as $\frac{g/h}{j/k}$, where $g$, $h$, $j$, and $k$ are all expressions generated by ``. According to Lemma 1, it's obvious that there exists a set $H' \subset H$ subsumes $\frac{gk}{jh}$.

If $s$ is a partial derivative, let $s = \partial e$. According to the inductive hypothesis, $e$ can be subsumed by a set $H_e \subset H$. According to the linearity of differentiation, we can assume $e$ is a single element of $H$. Without loss of generality, we assume $e$ is an expression generated by `<div>`. According to Lemma 4, the conclusion holds.

4. **Conclusion:** By the principle of mathematical induction, Theorem 1 holds.

### E.2 Proof of Theorem 2

*Proof.* Let $F_1^*$ be a subset of $F$ with cardinality $k_1$ that realizes the $k_1$-level margin, such that $\text{LevelMargin}_F(k_1) = \text{Margin}_F(F_1^*)$.

Since $k_1 < k_2 \le |F|$, we can construct a new set $F_2'$ by adding $k_2 - k_1$ arbitrary terms from $F \setminus F_1^*$ to $F_1^*$. By construction, $F_1^* \subseteq F_2'$ and $|F_2'| = k_2$.

The minimal residual is found by solving a linear least-squares problem. The solution space for the coefficients $\boldsymbol{\xi}$ corresponding to the terms in $F_1^*$ is a subspace of that for the terms in $F_2'$. Therefore, the minimal residual over the larger set $F_2'$ cannot be greater than the minimal residual over its subset $F_1^*$:

$$\|\text{Res}^*(\text{NN}_\theta, F_1^*)\|_2 \ge \|\text{Res}^*(\text{NN}_\theta, F_2')\|_2.$$

From the definition of Margin (Definition 3), this directly implies that:

$$\text{Margin}_F(F_1^*) \ge \text{Margin}_F(F_2').$$

Furthermore, by the definition of $k$-Level Margin (Definition 4), the $k_2$-level margin is the minimum margin over all subsets of size $k_2$. The margin of our constructed subset $F_2'$ must therefore be greater than or equal to this minimum:

$$\text{Margin}_F(F_2') \ge \text{LevelMargin}_F(k_2).$$

Combining these inequalities, we arrive at the desired result:

$$\text{LevelMargin}_F(k_1) = \text{Margin}_F(F_1^*) \ge \text{Margin}_F(F_2') \ge \text{LevelMargin}_F(k_2).$$

This completes the proof. $\qquad\qquad\square$

## F  More Experimental Details

### F.1  Equation Form

Table 3: Summary of Governing Equations

| Equation | Form |
| --- | --- |
| Burgers' Equation | $\partial_t u = -u\partial_x u + \left(\frac{0.01}{\pi}\right)\partial_{xx}u$ |
| Schrödinger Equation | $\begin{cases} \partial_t u = -0.5\partial_{xx}v - v(u^2 + v^2) \\ \partial_t v = 0.5\partial_{xx}u + u(u^2 + v^2) \end{cases}$ |
| Navier-Stokes Equation | $\begin{cases} \partial_t u = -u\partial_x u - v\partial_y u - \partial_x p + 0.01(\partial_{xx}u + \partial_{yy}u) \\ \partial_t v = -u\partial_x v - v\partial_y v - \partial_y p + 0.01(\partial_{xx}v + \partial_{yy}v) \end{cases}$ |

## F.2 L2 Relative Error

**Definition 7** (L2 Relative Error). Let $f = [f_1, f_2, \ldots, f_n]$ be a vector of true values and $\hat{f} = [\hat{f}_1, \hat{f}_2, \ldots, \hat{f}_n]$ be a vector of approximations. The L2 Relative Error (L2RE) is defined as:

$$\text{L2RE} = \frac{\|f - \hat{f}\|_2}{\|f\|_2} = \frac{\sqrt{\sum_{i=1}^{n} |f_i - \hat{f}_i|^2}}{\sqrt{\sum_{i=1}^{n} |f_i|^2}}.$$

Please note that for coefficient estimation, the L2RE is simplified as follows:

$$\sum_{i=1}^{n} \frac{|\xi_i - \hat{\xi}_i|}{|\xi_i|}.$$

The benefit of using this metric is that errors in large coefficients do not dominate and obscure errors in small coefficients.

## F.3 Network Architecture and Training Details

Our model's architecture is a multilayer perceptron (MLP) composed of 8 hidden layers. For the Burgers' Equation, each layer has 20 neurons. This architecture is adapted for the Schrödinger Equation by modifying the output layer to predict two variables (the real and imaginary parts of the solution). For the more complex Navier-Stokes Equations, we increase the network's width to 40 neurons per hidden layer.

The training data consists of $1 \times 10^4$ (40%), $2 \times 10^4$ (40%), and $2 \times 10^5$ (20%) measurement points for the Burgers', Schrödinger, and Navier-Stokes experiments, respectively. We hold out 20% of the training data as a validation set, which is used to select the derivative computation weights and the hyperparameter $\lambda$. The networks are pre-trained for 5,000 epochs for the Burgers' and Schrödinger equations, and 100,000 epochs for the Navier-Stokes equations. Across all tasks, we use a consistent initial sparsity of 10 and a margin threshold of 0.05. During consistency optimization, MSS is applied every 1,000 epochs to refine the discovered equation. The values for $\lambda$ are selected based on validation error and are summarized in Table 4. All experiments were conducted on a server equipped with four NVIDIA A6000 GPUs, each with 48 GB of memory.

Table 4: Hyperparameter $\lambda$ settings.

| Method | Burgers' Equation | Schrödinger Equation | Navier-Stokes Equations |
|---|---|---|---|
| ABL-PDE (w/o FEM) | 0.05 | 0.5 | 1 |
| ABL-PDE | 5 | 5 | 200 |

# G  Detailed Experimental Results

## G.1 Redundant Terms

Table 5 details the specific redundant terms for the dagger-marked (†) results presented in Table 2.

Table 5: Redundant terms of the dagger-marked results in Table 2.

| Method | Equation | Noise Level | Redundant Terms |
|---|---|---|---|
| ABL-PDE (w/o CO) | Schrödinger Eq. | All Levels | $u_t$: $u_{xx}, u, uv, u^3, v$
$v_t$: $-$ |
| DL-PDE++ | Navier-Stokes Eq. | 0% | $u_t$: $-$
$v_t$: $v_x$ |
| | | 5% | $u_t$: $p^2u, vv_x$
$v_t$: $p_yu, pu_y$ |
| | | 10% | $u_t$: $p^2u, vv_x, p^3, p^2$
$v_t$: $p_yu, pu_y, v_x$ |

## G.2 Test Error

Table 6 presents the detailed results of test errors on three benchmarks. The area of the shaded circles in Figure 3 is proportional to the square root of the standard deviation. We use the square root to improve the visibility of the standard deviation of ABL-PDE.

Table 6: Test error comparison results on three benchmarks. The evaluation metric is the sum of L2RE (%) on each dependent variable. Results of ABL-PDE (w/o FEM) and ABL-PDE are the mean and standard deviation of 5 experiments with different hyperparameters.

| Method | Burgers' Equation | | | Schrödinger Equation | | | Naiver-Stokes Equation | | |
|---|---|---|---|---|---|---|---|---|---|
| | 0 | 5% | 10% | 0 | 5% | 10% | 0 | 5% | 10% |
| ABL-PDE (w/o CO) | 1.79 | 1.83 | 1.92 | 2.25 | 2.37 | 3.93 | 2.85 | 3.38 | 4.70 |
| ABL-PDE (w/o FEM) | $1.04 \pm 0.11$ | $1.06 \pm 0.11$ | $1.08 \pm 0.19$ | $1.67 \pm 0.40$ | $1.74 \pm 0.38$ | $1.84 \pm 0.36$ | $3.06 \pm 0.22$ | $3.44 \pm 0.25$ | $4.02 \pm 0.24$ |
| ABL-PDE | $\mathbf{0.68 \pm 0.01}$ | $\mathbf{0.72 \pm 0.02}$ | $\mathbf{0.85 \pm 0.03}$ | $\mathbf{1.22 \pm 0.01}$ | $\mathbf{1.36 \pm 0.01}$ | $\mathbf{1.75 \pm 0.01}$ | $\mathbf{2.64 \pm 0.05}$ | $\mathbf{3.17 \pm 0.04}$ | $\mathbf{3.88 \pm 0.06}$ |

## G.3 Redundancy Reduction

Table 7 presents the detailed number of candidate terms.

Table 7: Number of candidate terms w.r.t. operator number limit.

| Number | 2 | 3 | 4 | 5 | 6 | 7 | 8 | 9 | 10 |
|---|---|---|---|---|---|---|---|---|---|
| Full | 105 | 44205 | 7.8E+9 | 2.4E+20 | 2.4E+41 | 2.3E+83 | 2.1E+167 | 1.7E+335 | 1.2E+671 |
| Canonical | 39 | 143 | 510 | 1784 | 6141 | 20849 | 69948 | 234789 | 764761 |
| ABL-PDE | 11 | 29 | 69 | 165 | 387 | 879 | 1955 | 4254 | 9053 |

