# OpenReview forum: "Discovering Symbolic Partial Differential Equation by Abductive Learning"
_NeurIPS.cc/2025/Conference — NeurIPS 2025 poster_

### Official Review · Reviewer_B51F · 2025-06-30

**Clarity:** 3
**Significance:** 3
**Originality:** 4
**Rating:** 5
**Confidence:** 4

**Summary:**

This paper presents a methodology for discovering the differential operator of Partial Differential Equations using Abductive Learning. The method is structured in two steps, first a neural network is used to approximate the given data such that there is no need for an integrator and second a process that contains an outer and an inner loops is considered. The outer loop increases the length of the differential operator, the inner loop picks the candidate operators, use Mean Subset Selection to find the shorted expression and fine-tune the neural network. The process is performed until convergence on a level margin criterion. The method is tested on three PDEs for different levels of noise against different baselines, and on two PDEs specifically against D-CIPHER which cannot handle systems.

**Questions:**

- Could you provide more information about how the method handles additive coefficients?

- When performing re-writes of complex terms in the grammar, is there a way to derive a bound to this growth such the method remains efficient when considering more complex expressions (ones with forcing terms for example).

-  Maybe I missed it but could you clarify if the candidate terms are the same for all PDEs or not? And if not how they are chosen?

- How many times did you run D-CIPHER and what was the best result?

- Why is the discovery more accurate for 5% error?

**Ethical Concerns:**

["NO or VERY MINOR ethics concerns only"]

**Final Justification:**

The reviewers answered my questions and I am satisfied with the responses.

**Limitations:**

There is only one limitation stated in the conclusions. I believe that the authors should include bullets from the weaknesses section to the limitations.

**Paper Formatting Concerns:**

The format looks fine to me.

**Quality:**

3

**Strengths And Weaknesses:**

Strengths:

- This is an interesting method that combines canonical grammars that are expressive enough, see Theorem 1, together with a consistency optimization where is showed that the k-level margin decreases monotonically. The method essentially proposes the idea of reducing the hypothesis using a first-order logic knowledge base and integrity constraints, and delivers state-of-the-art accuracy for PDEs with different levels of noise.

Overall, it provides a novel method to cut down the hypothesis space of combinatorial optimization methods, which is very valuable. Also the paper is mostly well structured, well written, and the authors provide helpful information in the appendix.

Weaknesses:

Clarity: I believe that there are some parts that are not completely clear to me. For example:

- How are constants treated in the Grammar. The non-terminal <unit> allows for (in-)dependent variables, and partials only, meaning that constants, for example a source term "+2", cannot be generated. However, constants appear in the target equations, for example 0.01/pi in the Burger's. In my understanding, the authors rely on the \xi vector in the FEM training to account for that but this doesn't work for additive constants, such as u_t - D u_xx - S where S a constant source.

- There is the following subtle issue in the method (at least to my understanding): The strength of using a grammar is to keep the library of candidate term small such that the MSS choice remains tractable. Theorem 1 states that any expression, even if it is nested, can be re-written as linear combination of terms from the canonical grammar. However, when this happens a single complex node is substituted by multiple simpler nodes, which means that you might lose the benefit of restricting the space of candidate expressions and having a large library again.

- I believe that there is one more subtle issue with Theorem 1. In my understanding, Theorem 1 guarantees equivalence of symbolic expressions, but does not guarantee that those equivalences hold as PDE operators acting on a solution space with the correct boundary conditions. The difference is that without discussing boundary condition assumptions, probably smoothness also because maybe the solution has weak derivatives, you can be sure that the expansion holds in the weak form where you consider in the FEM residual.

- In my understanding, Theorem 1 assumes that derivatives and variables commute and the grammar contains the basic terms, but it seems like constants are missing from the grammar.

- It is not clear how the FEM based training handles boundary conditions in the weak formulation. It seems like the training ignores what happens on the boundary. Moreover, I believe that there should be some discussion on how the conditioning of the mass matrix affects the overall  training, as M might be ill-conditioned for coarse meshes.

- It is not clear to me how expensive and numerically stable is the LevelMargin step as it needs to compute orthogonal projections which is expensive and might be numerically unstable for colinear terms.

Lemma 5:

- In the proof it is stated that "Without a loss of generality we assume $v \perp Y$.". This statement is not justified and the derivations that follow depend critically on that orthogonality. This statement is not without a loss of generality because if v has a component inside Y, replacing by its orthogonal component v_{ortho} = v - Proj_Y v changes the left-hand side of the inequality because the projection onto span(v) does not coincide with the projection on span(v_{ortho}).  This means that the inequality in Theorem 2 does not necessarily hold

. If the authors impose the condition $v \perp Y$, the inequality is true but the method does not apply to arbitrary grammars, because you would need to guarantee the orthogonality.

- The results for D-CIPHER are (in my understanding) in contradiction with the original paper, that shows that D-CIPHER is robust to noise.

---

> ### Author Rebuttal · Authors · 2025-07-29
>
> We thank Reviewer B51F for the insightful review, the careful examination of our paper, and the appreciation of the value and innovation of our work. Below, we answer your questions in detail, hoping to address your concerns.
>
> ---
>
> **Q1.** Clarification on constants in the grammar and additive coefficients.
>
> **A1.** Thanks for your question. In this work, the only constants are the linear combination coefficients in Equation (1). These constants are not generated when abducing candidate terms; instead, they are introduced during FEMTrain as trainable parameters. To introduce a constant source, it suffices to add an additional column of ones and assign it a trainable coefficient. We will revise Equation (1) to explicitly include the additive coefficient for clarity.
>
> ---
>
> **Q2.** Concern regarding the re-write of complex terms.
>
> **A2.** Thanks for your question. The growth rate in the number of terms caused by re-writing is indeed an interesting question. While deriving a formal bound is difficult due to the lack of a clear definition for a *compact term*, we can offer an intuitive explanation.
>
> First, it is more appropriate to consider rewriting sets of complex expressions, since a PDE typically contains multiple terms. Consider a set of complex expressions A and a set of simpler, canonical expressions B that can subsume A. Since the full space of complex expressions contains exponentially more elements than the space of canonical terms, it is clear that the size of B does not necessarily grow exponentially with the size of A; otherwise the full space of simpler canonical terms would be larger.
> Moreover, we generally prefer an increase in the number of terms rather than an increase in the number of symbols per term. This is because the search space grows exponentially with the number of symbols, whereas increasing the number of terms does not expand the full search space of MSS, which is desirable for our approach. We will add this discussion to the next version of our paper.
>
> ---
>
> **Q3.** Clarification on the FEM-based residual and operator equivalence in Theorem 1.
>
> **A3.** Thank you for your insightful comment. We recognize that our use of the term 'weak formulation' was misleading, for which we sincerely apologize. You are correct that our FEM-based residual does not contain the boundary terms expected from a traditional weak formulation.
>
> To clarify, our FEM-based residual is more accurately described as a **Galerkin projection of the strong-form residual**. We compute derivative terms (e.g., $u_{xx}$) directly using automatic differentiation on a neural network, and then project the resulting strong-form PDE residual onto the FEM basis functions. Because this procedure does not involve integration by parts, boundary integrals do not arise in this part of our loss function. The boundary values of the solution are enforced via the MSE component of the consistency measure defined in Equation (2). We will revise the manuscript to reflect this precise description of our approach.
>
> Regarding your other concern on Theorem 1, you are correct that its application as PDE operators relies on certain assumptions. In our framework, the crucial assumption is that the solution is sufficiently **smooth** for the required derivatives to exist. This smoothness is provided by the neural network, which acts as a continuous and differentiable representation of the solution field. The derivatives are then evaluated via automatic differentiation. We will clarify in our revision that the practical application of Theorem 1 rests on this smoothness being effectively captured by the neural network backbone.
>
> Thank you again for your valuable feedback, which will significantly improve the clarity and accuracy of our paper. Your insight is also very enlightening for us. We consider the development of a solver capable of automatically performing symbolic weak form computations, as well as methods for the adaptive selection of finite element basis functions, to be important directions for our future work.
>
> ---
>
> **Q4.** Clarification on the computational cost and numerical stability of the LevelMargin step.
>
> **A4.** Thanks for your question. The projection step in MSS is a standard least-squares problem, and the dimension of the subspace projected onto each time does not exceed the initial sparsity. Since there is a closed-form solution, the computation is very efficient. In our implementation, we use PyTorch’s method for computing matrix pseudo-inverse, which robustly handles colinear terms. We did not observe any numerical instability in our experiments.
> As for the condition number of the mass matrix M, we do not need to be concerned about M being ill-conditioned. This is because we are free to choose the mesh for the FEM-based residual in the consistency measure. "Certainly, the effect of the condition number of M on discovery performance is a valuable question. We consider this an important point for our subsequent research.
>
> ---
>
> **Q5.** Clarification on the proof of Lemma 5.
>
> **A5.** Thank you very much for the detailed review! You clearly pointed out that when $v$ is not orthogonal to $Y$, the left side of the inequality will change. However, please note that this does not cause the inequality to fail. We omitted some details in the original proof, which we now provide in full.
>
> **Proof:**
>
> Let $v$ be a vector that is not orthogonal to the space $\text{span}(Y)$. Decompose $v$ into two orthogonal components: $v = v\_Y + v\_{Y^\perp}$, where $v\_Y = \text{Proj}_{\text{span}(Y)}(v)$ is the component of $v$ in $\text{span}(Y)$, and $v\_{Y^\perp}$ is the component orthogonal to $\text{span}(Y)$.
>
> Since $\text{span}(Y \cup \{v\}) = \text{span}(Y \cup \{v\_{Y^\perp}\})$, the RHS remains unchanged:
> $$
> \\|\text{Proj}\_{\text{span}(Y \cup \{v\})}(u)\\|\_2 - \\|\text{Proj}\_{\text{span}(Y)}(u)\\|\_2 = \\|\text{Proj}\_{\text{span}(Y \cup \{v\_{Y^\perp}\})}(u)\\|\_2 - \\|\text{Proj}\_{\text{span}(Y)}(u)\\|\_2.
> $$
>
> For the LHS, consider the relationship between $v_Y$ and $\text{span}(X)$:
>
> 1. **Case 1:** $v_Y \in \text{span}(X)$
>    If $v_Y$ lies in $\text{span}(X)$, then $\text{span}(X \cup \{v\}) = \text{span}(X \cup \{v_{Y^\perp}\})$. The inequality becomes:
>    $$
>    \\|\text{Proj}\_{\text{span}(X \cup \{v\_{Y^\perp}\})}(u)\\|\_2 - \\|\text{Proj}\_{\text{span}(X)}(u)\\|\_2 \ge \\|\text{Proj}\_{\text{span}(Y \cup \{v\_{Y^\perp}\})}(u)\\|\_2 - \\|\text{Proj}\_{\text{span}(Y)}(u)\\|\_2.
>    $$
> 2. **Case 2:** $v_Y \notin \text{span}(X)$
>    Decompose $v_Y$ further as $v_Y = v\_{Y,X} + v\_{Y,X^\perp}$, where $v\_{Y,X} = \text{Proj}_{\text{span}(X)}(v_Y)$ and $v\_{Y,X^\perp}$ is orthogonal to $\text{span}(X)$. Then:
>    $$
>    v = v\_{Y^\perp} + v\_{Y,X} + v\_{Y,X^\perp}.
>    $$
>    The subspace $\text{span}(X \cup \{v\})$ simplifies to $\text{span}(X \cup \{v\_{Y^\perp}, v\_{Y,X^\perp}\})$. According to the Pythagorean theorem, projecting onto a larger subspace increases the norm and therefore the following inequality holds:
>    $$
>    \\|\text{Proj}\_{\text{span}(X \cup \{v\})}(u)\\|\_2 \ge \\|\text{Proj}\_{\text{span}(X \cup \{v\_{Y^\perp}\})}(u)\\|\_2.
>    $$
>
> From Lemma 5:
> $$
> \\|\text{Proj}\_{\text{span}(X \cup \{v\_{Y^\perp}\})}(u)\\|\_2 - \\|\text{Proj}\_{\text{span}(X)}(u)\\|\_2 \ge \\|\text{Proj}\_{\text{span}(Y \cup \{v\_{Y^\perp}\})}(u)\\|\_2 - \\|\text{Proj}\_{\text{span}(Y)}(u)\\|\_2.
> $$
>
> Combining results, we have:
> $$
> \\|\text{Proj}\_{\text{span}(X \cup \{v\})}(u)\\|\_2 - \\|\text{Proj}\_{\text{span}(X)}(u)\\|\_2 \ge \\|\text{Proj}\_{\text{span}(Y \cup \{v\})}(u)\\|\_2 - \\|\text{Proj}\_{\text{span}(Y)}(u)\\|\_2.
> $$
>
> This confirms the inequality in all cases.
>
> ---
>
> **Q6.** Clarification on the comparison with D-CIPHER.
>
> **A6.** Thanks for your question. D-CIPHER is a SOTA PDE discovery method.
> One of its characteristics is to integrate measurements of the same PDE under multiple initial and boundary conditions, which likely contributes to its robustness to noise. However, since our benchmarks do not involve multiple datasets for one PDE, we set the number of datasets to 1 for D-CIPHER to align the experimental setup and repeated each experiment 5 times, reporting the best result. To address your concerns about the experimental results, we have re-run D-CIPHER using the original paper’s hyperparameter settings (10 datasets, 10 trials) and report the results using their metric, RMSE, as shown in the table below.
>
> ||Burgers' Equation|||Heat Equation|||
> |---|---|---|---|---|---|---|
> ||0|5%|10%|0|5%|10%|
> |D-CIPHER|1.61e-2|8.13e-2|1.49e-1|4.37e-4|1.87e-2|2.62e-2|
> |ABL-PDE|**1.77e-3**|**4.17e-3**|**7.11e-3**|**4.04e-4**|**5.03e-5**|**1.17e-3**|
>
> The experimental results align well with those reported in Figure 3 of the original paper. We will include these updated results and provide a clearer description of the experimental setup in the next version of the paper.
>
> ---
>
> **Q7.** Clarification on the candidate term selection.
>
> **A7.** We use the same knowledge base to abduce candidate terms for all tasks, with the only difference being that we state different dependent and independent variables for each task.
>
> ---
>
> **Q8.** Clarification on the discovery result under 5% noise.
>
> **A8.**
> Thank you for your detailed review of our paper! This non-monotonic result likely arises from our consistency optimization (CO) stage for two reasons:
> 1.  **Hyperparameter $\lambda$:** We used the same $\lambda$ across different noise levels. It's plausible that the chosen $\lambda$ is slightly better suited for a 5% noise measurement.
> 2.  **Noise as Regularization:** A small amount of noise can act as a regularizer, preventing overfitting and improving the model's generalization. Better generalization is critical for discovering a generalizable PDE, sometimes more so than perfectly fitting clean data.
>
> Notably, the physical field fitting error shown in our paper's Figure 3 remains monotonic with noise, which supports the overall validity of our experiments.

---

> > ### Comment · Reviewer_B51F · 2025-08-04
> >
> > I thank the reviewers for the comprehensive answers. I would still like two clarifications.
> >
> > About Q2: i) You are correct that, in a counting sense the set of all possible canonical terms of size \leq n is smaller than the set of all terms of the same symbol number. However, this fact alone does not bound a potential blow up in size, e.g. for some A the expansion  A \mapsto B can still be exponential in A.
> > ii) In any subset-selection procedure, the search space of models is a power set of the library. If the library grows from p to \Delta p, the number of possible subsets grows from 2^p to 2^{p + \Delta p}. So, adding terms does enlarge the space. Maybe what you mean is that your heuristic does not enumerate all 2^{p + \Delta p} subsets so it is not affected by a lot, but this is an algorithmic property, not an intrinsic property of the library size.
> > So, your answer relies on two claims: i) rewriting never creates exponentially many canonical terms relative to the original set, and ii) enlarging the library does not affect the MMS search space, that are not quantified somehow such that you can claim that the blow-up is manageable in total.
> >
> > In my opinion, you should acknowledge the fact and add it to the limitations.
> >
> > About Q5, Case 2: Maybe I am mistaken (I am a little rusty in linear-algebra), but your inequality $\| Proj_{\span(X \cup \{ v\})} u \| \geq \| Proj_{\span(X \cup \{ v_{Y^\perp}\})} u \| does not hold. Projection norm monotonicity applies when one subspace is contained in the other. Here the two subspaces are typically incomparable. So, in my understanding your Lemma doesn't hold.
> >
> > Can you please provide more information about why your process is correct or maybe provide an argument that does not rely on subspace inclusion?

---

> > > ### Author Response · Authors · 2025-08-04
> > >
> > > Thank you for your thorough and insightful review.
> > >
> > > Regarding Q2, there are indeed expression terms that, after being rewritten, produce an exponentially large number of canonical terms. You are also correct that the intrinsic size of the search space differs from the size of the space after heuristic pruning. We will add a discussion of this to our revised version.
> > >
> > > Regarding Q5, you are correct that Lemma 5 does not hold in Case 2. We mistakenly assumed the projection onto $v_{Y^\perp}+v_{Y,X^\perp}$ was equivalent to the projection onto $\{v_{Y^\perp}\} \cup \{v_{Y,X^\perp}\}$. Accordingly, we have performed a preliminary test on the Burgers' Equation and found that this modification did not affect our results. We will correct the calculation of the Level Margin in Algorithm 1 and report the new experimental results in the next version of our paper.
> > >
> > > Thank you again for your insightful comments. They have been immensely helpful in improving the clarity and soundness of our paper.

---

> > > > ### Comment · Reviewer_B51F · 2025-08-05
> > > >
> > > > Thank you for your replies and for clarifying my questions.
> > > >
> > > > Regarding Q5
> > > > Your Burgers experiment shows that the method works in practice, however the proof of monotonic level margin still relies on the projection norm monotonicity between two subspaces that are not nested. So, the current argument does not guarantee that the margin necessarily decreases at each step, so Theorem 2 does not hold for the general case.
> > > >
> > > > You could fix this (if I am not mistaken) by keeping the margin definition as is, but explicitly add the orthogonality requirement $v \perp Y$ to Lemma 5 and Theorem 2. In general this is a strong assumptions, that is almost never the case in practice unless you keep an orthogonal basis (orthogonalize with respect to the current span), as for example in orthogonal marching pursuit. So, for $v \perp Y$  to work it requires that your algorithm orthogonalises each candidate term before computing the margin.

---

> > > > > ### Author Response · Authors · 2025-08-07
> > > > >
> > > > > We sincerely appreciate your suggestion. We completely agree that the assumption of $v$ being orthogonal to $Y$ is excessively strong. Therefore, we sought an alternative metric to control for sparsity. Inspired by the Orthogonal Matching Pursuit (OMP) algorithm you mentioned, we have revised the definition of the margin as follows:
> > > > >
> > > > > **Definition (Margin):** Let $H$ be a set of expression terms and $F \subseteq H$. The margin of $F$ with respect to $H$ is defined as:
> > > > >
> > > > > $$
> > > > > \text{Margin}\_{H}(F) = \frac{\||\text{Res}(\text{NN}\_{\theta}, F)\||_2 - \||\text{Res}(\text{NN}\_{\theta}, H)\||_2}{\||\text{Res}(\text{NN}\_{\theta}, H)\||_2}
> > > > > $$
> > > > >
> > > > > where $\text{Res}(\text{NN}\_{\theta}, \cdot)$ is the residual obtained after projecting $\partial_t \text{NN}\_{\theta}$ onto the subspace spanned by the second argument.
> > > > >
> > > > > The definition of the k-Level Margin remains unchanged. It can be easily proven that for a given set $H$, its k-Level Margin is monotonically non-increasing as $k$ increases. Consequently, we can control the sparsity with level margin. Specifically, we search for the smallest $k$ such that the k-Level Margin is less than a margin threshold.
> > > > >
> > > > > We use the set filtered by POSS as the universal set $H$, set the margin threshold to 0.05, and reran the experiments for ABL-PDE(w/o CO) and ABL-PDE. The parts that differ from Table 2 in the main text are presented in the table below:
> > > > >
> > > > > |  | Burgers' Equation | Schrodinger Equation |  |  | Navier Stokes Equation |
> > > > > |---|---|---|---|---|---|
> > > > > |  | 10% | 0 | 5% | 10% | 10% |
> > > > > | ABL-PDE(w/o CO) | 25.14 | 14.11 | 15.77 | 15.44$^&dagger;$ | 63.31$^&dagger;$ |
> > > > > | ABL-PDE | **4.926** | **2.218** | **2.845** | **3.013** | **37.04** |
> > > > >
> > > > > As shown in the table, the new sparsity control metric is effective. Initial redundant terms only appear in the cases of the Schrödinger Equation with 10% noise ($u$) and the Navier-Stokes Equation with 10% noise ($\partial_x v$), and these terms are subsequently eliminated during the consistency optimization stage.
> > > > >
> > > > > We will update the results in the next version of the manuscript after completing experiments with different $\lambda$ values. Thank you again for your invaluable review; your suggestions have significantly improved the quality of our paper.

---

> ### Comment · Reviewer_B51F · 2025-08-07
>
> Thank you for redesigning the margin. This removes the orthogonality issue nicely.
>
> Now you just need to do a few minor things to finalize the revision i) restate Lemma 5/ Theorem 2 under the new definition and provide a short proof that making $F \subseteq H$ can only reduce $ || \text{Res} ||_2$, ii) provide a short clarification on how the fixed library $H$ is chosen and how sensitive is the method to the $0.05$ threshold, iii) comment on whether OMP-style guarantees apply to your method or not.
>
> With these additions all my concerns about the soundness of the theoretical part will be fully resolved and I will increase score to acceptance.

---

> > ### Author Response · Authors · 2025-08-08
> >
> > Thank you very much for your detailed and constructive suggestions. We will revise our paper accordingly.
> >
> > 1.  We will revise the definition of the margin and update Lemma 5 and Theorem 2 accordingly. The proof is straightforward, following from a combination of subspace inclusion and the Pythagorean theorem.
> >
> > 2.  We will clarify in the paper that H is the set filtered by POSS. Regarding the margin threshold, we favor a smaller threshold that tries to include all potentially relevant terms when the MSS algorithm is first applied. This is because, empirically, it is easier to remove a false positive term than to add a false negative one.
> >
> > 3.  To find a suitable sparsity control metric, we experimented with the OMP algorithm from the scikit-learn library. However, it proved ineffective for our purpose because its termination condition is limited to either the number of terms or the absolute norm of the residual. We plan to manually implement the OMP algorithm and adopt a relative residual, similar to our margin definition, as the stopping criterion. When the quality of H output by POSS is high (i.e., the correlation between different terms is low) and the underlying equation is sufficiently sparse, the incoherence condition for perfect recovery is met. In such cases, the OMP algorithm alone could efficiently identify the form of the equation.
> >
> > We will incorporate these discussions into the revised manuscript. Thank you again for your valuable review!

---

> > > ### Comment · Reviewer_B51F · 2025-08-08
> > >
> > > I raised my score. Thanks for the nice work!

---

> > > > ### Author Response · Authors · 2025-08-08
> > > >
> > > > Thank you very much for your encouraging comments and for raising the score. We are grateful for your time and thoughtful review of our manuscript. Thanks!

---

### Official Review · Reviewer_dxqN · 2025-07-02

**Clarity:** 3
**Significance:** 2
**Originality:** 3
**Rating:** 3
**Confidence:** 4

**Summary:**

This paper introduces ABL-PDE, a novel abductive learning-based approach for PDE discovery. The key contribution is the transformation of the traditional unidirectional discovery pipeline into a bidirectional enhancement loop, improving both structure and parameter identification. Specifically, the method incorporates FOL KB to represent a broad class of PDEs with reduced redundancy and enhanced interpretability. Extensive experiments across multiple PDE tasks and noise levels demonstrate that ABL-PDE achieves state-of-the-art performance in both equation structure recovery and coefficient estimation.

**Questions:**

1. In Table 2, the authors compare the performance of ABL-PDE and its variants with the baselines PDERidge and DL-PDE++. I have a few questions and comments:
(1) I noticed that some L2RE values are exactly the same across different methods for the same PDE and noise level. Is this because the same Ridge regression is used in the final step of those methods?
(2) Why are standard deviations (± values) reported for some results but not others? Is this because those experiments are deterministic, with no variation across runs?
(3) I understand that the dagger symbol indicates the discovered PDE includes redundant terms. This is common in symbolic regression. However, how severe are these cases? For example, one extra term with a tiny coefficient might be fine, but having five extra terms with meaningful coefficients in a three-term PDE could be considered a failure. Also, does the dagger indicate that all correct terms are present and only redundant ones are added, with nothing missing?
(4) If an extra term is included in the discovered PDE, how does the L2RE be calculated? Do you ignore the coefficients of redundant terms and only use the correct terms when computing the error?

2. Could the authors compare their method and its variants with more recent and representative baselines? PDERidge only applies Ridge regression on a known PDE structure, which is not a complete PDE discovery method. DL-PDE++ (2021) is somewhat outdated, though still acceptable. It would be helpful to include comparisons with more modern methods such as Symbolic Physics Learner (Sun et al., ICLR 2023) or SGA-PDE (Chen et al., Physical Review Research 2022).

3. Since the overall optimization process alternates between MSS and FEMTrain, can the authors provide a proof or argument that this process always converges?

4. The caption of Figure 3 mentions that 5 different values of lambda (used in Equation 2) are considered. However, neither Figure 3 nor Appendix G clearly shows or discusses these different lambda values. Could the authors clarify this?

5. In section 6.4, it is stated that the lambda values used are [0.5λ, 0.75λ, λ, 1.25λ, 1.5λ]. Does this mean five separate configurations with λ set to 0.5, 0.75, 1.0, 1.25, and 1.5 respectively? The wording is a bit unclear.

**Ethical Concerns:**

["NO or VERY MINOR ethics concerns only"]

**Final Justification:**

I have read the author rebuttal and the discussions among all reviewers. The authors have addressed my concern regarding hyper-parameter settings. However, my core concern remains - the choice of reporting only the L2RE of term coefficients as the sole experimental metric is insufficient. I recommend also reporting the recovery rate (success rate) together with the equation value MSE (or RMSE, or any MSE-based metric) calculated on the entire equation, not just the coefficients.

The quality of equation discovery depends both on the proportion of correct terms recovered and on the accuracy of the predicted values. This is also why the reported performance improvement of ABL-PDE over the baselines remains unclear to me (e.g., baseline performance comparison in A2), as the chosen metric is not well-suited for a comprehensive evaluation. For the baseline experiments mentioned in all reviewers' comments, if available, please include results using the updated metrics in the revised version.

Furthermore, as noted in Q3 and A3, a convergence proof between MSS and FEMTrain is necessary to establish a solid foundation for ABL-PDE. Taking these points into account, I will maintain my rating for this paper.

**Limitations:**

Yes.

**Paper Formatting Concerns:**

Not applied.

**Quality:**

3

**Strengths And Weaknesses:**

### Strengths
1. The authors replaces the traditional unidirectional pipeline with a feedback loop between symbolic and numerical components, enhancing robustness and accuracy.
2. The authors introduce a knowledge base to constrain the candidate search space, effectively reducing the problem size to a computationally manageable scale. This approach is both practical and innovative - I like this idea.

### Weaknesses
1. The Experiments section includes a limited number of baseline methods for comparison. The method could benefit from comparisons with more recent or diverse PDE discovery methods.
2. The alternating optimization between MSS and FEMTrain lacks a theoretical guarantee of convergence, which may affect reproducibility or stability in some scenarios.
3. The choice of using L2RE of term coefficients can not clearly reflect the performance of equation discovery.

---

> ### Author Rebuttal · Authors · 2025-07-29
>
> We thank Reviewer dxqN for detailed review, helpful comments, and appreciation on the practicality and innovation of our work. Below, we answer your questions in detail.
>
> ---
>
> **Q1.** "In Table 2, the authors compare the performance of ABL-PDE and its variants with the baselines PDERidge and DL-PDE++. I have a few questions and comments: ... Do you ignore the coefficients of redundant terms and only use the correct terms when computing the error?"
>
> **A1.**
> Thanks for your helpful question. We will revise our paper to clarify these points.
> 1. Exactly. We used the same pre-trained neural network and the same regression method to ensure a fair comparison.
> 2. The standard deviation presented here is with respect to our method's hyperparameter $\lambda$, which is not applicable to other methods. As mentioned in Section 6.2, results for DL-PDE++ were obtained through careful tuning of its hyperparameters. When we are unable to tune its hyperparameters to discover the correct equation, we ensure it includes all ground truth terms and minimize redundant ones. For our method, we try to include relevant terms as much as possible at the outset by setting a very small margin threshold ($1 \times 10^{-3}$), and redundant terms will then be eliminated later through consistency optimization.
> 3. Thanks for your comments. We list the number of redundant terms below and will add this to the next version of our paper (empty cells indicate no redundant terms):
>
> |Method|Burgers' Equation|Schrödinger Equation|||Navier-Stokes Equations|||
> |:-:|:-:|:-:|:-:|:-:|:-:|:-:|:-:|
> ||10%|0|5%|10%|0|5%|10%|
> |DL-PDE++|||||1|4|7|
> |ABL-PDE (w/o CO)|2|3|3|3||||
>
> 4. Exactly. When computing the error, we ignore the coefficients of redundant terms and only use the correct terms.
>
> ---
>
> **Q2.** "Could the authors compare their method and its variants with more recent and representative baselines? ... It would be helpful to include comparisons with more modern methods such as Symbolic Physics Learner (Sun et al., ICLR 2023) or SGA-PDE (Chen et al., Physical Review Research 2022)."
>
> **A2.** Thanks for your suggestion. These are indeed two highly influential works. We extensively discussed SGA-PDE in our paper, treating it as a representative of asynchronous methods. While Symbolic Physics Learner (SPL) doesn't directly involve PDEs, its core idea of using MCTS to handle large search spaces is orthogonal to our approach and offers significant inspiration for our future work. We will add a discussion of SPL in the next version of our paper.
> Since the code for SPL only covers ODEs rather than PDEs, applying it to our benchmark is non-trivial. Therefore, we invested effort to adapt ABL-PDE to ODEs and conducted experiments on their benchmark, Lorenz system.
> Similarly, because the code for SGA-PDE does not support the discovery of PDE systems, while 2 out of 3 of our benchmarks are systems of PDEs, we also applied ABL-PDE to the benchmarks used by SGA-PDE.
> We summarize the results in the table below. The results of SGA-PDE and SPL are taken from their respective papers. The results of our methods are obtained with the **same** initial sparsity $10$ and margin threshold $1 \times 10^{-3}$ as in the paper.
> Due to the tight rebuttal schedule, we did not select $\lambda$ based on the fitting error of the physical field on the validation set. Instead, we simply used a trick to rescale the supervised loss and the physics loss to similar magnitudes for **all the experiments**: $\text{loss} = \text{loss}_1 + \frac{\text{loss}_2}{\text{loss}_2 / \text{loss}_1 + 1e-8}$.
>
> |  | Burgers' Equation | KdV Equation | Chafee-Infante Equation |
> |---|---|---|---|
> | Ground Truth | $u_{t} = −uu_{x} + 0.1 u_{xx}$ | $u_{t}=-0.0025u_{xxx}-uu_{x}$ | $u_{t}=u_{xx}-u+u^3$ |
> | SGA-PDE[1] | $u_{t} = −1.0011 uu_{x} + 0.1024 u_{xx}$ | $u_{t}=-0.0025u_{xxx}-1.0004uu_{x}$ | $u_{t}=1.0002u_{xx}-1.0008u+1.0004u^3$ |
> | ABL-PDE (w/o CO) | $u_{t} = -0.83uu_{x}+0.096u_{xx}-0.026u+0.038 u^3$ | $u_{t}=-0.00237u_{xxx}-0.988uu_{x}$ | $u_{t}=0.843u_{xx}-0.852u+0.916u^3-0.0619uu_{x}+0.182u_{x}$ |
> | ABL-PDE | $u_{t} = −0.999uu_{x} + 0.10002 u_{xx}$ | $u_{t}=-0.00249u_{xxx}-0.995uu_{x}$ | $u_{t}=0.994u_{xx}-0.987u+0.996u^3$ |
>
> |  | Lorenz System |
> |---|---|
> | Ground Truth | $\dot{x} = −10x+10y$ &nbsp; $\dot{y} = 28x-y-xz$ &nbsp; $\dot{z} = -\frac{8}{3}z+xy$ |
> | SPL[2] | $\dot{x} = -9.966x + 9.964y$ &nbsp; $\dot{y} = 27.764x - 0.942y - 0.994xz$ &nbsp; $\dot{z} = -2.655z + 0.996xy$ |
> | ABL-PDE (w/o CO) | $\dot{x} = -9.807x+9.956y+0.0323xy-0.0185yz$ &nbsp; $\dot{y} = 27.72x - 0.976y - 0.989xz$ &nbsp; $\dot{z} = -2.698z + 0.950xy+0.0238yz$ |
> | ABL-PDE | $\dot{x} = -10.003x+10.002y$ &nbsp; $\dot{y} = 27.869x - 0.976y - 0.995xz$ &nbsp; $\dot{z} = -2.667z + 0.999xy$ |
>
> References:
>
> [1] Symbolic genetic algorithm for discovering open-form partial differential equations (SGA-PDE). Physical Review Research, 2022.
>
> [2] Symbolic Physics Learner: Discovering governing equations via Monte Carlo tree search. ICLR, 2023.
>
> ---
>
> **Q3.** "Since the overall optimization process alternates between MSS and FEMTrain, can the authors provide a proof or argument that this process always converges?"
>
> **A3.**
> Thanks for your question. A formal convergence proof is indeed challenging, since the overall process alternates between two complex subproblems: structure identification (MSS) and parameter/coefficient optimization (FEMTrain). Proving convergence for such alternating schemes is notoriously difficult, especially when the subproblems are not convex, which is the case here. For future work, we will consider analyzing the convergence of our method under reasonable assumptions.
>
> ---
>
> **Q4 & Q5.** "The caption of Figure 3 mentions that 5 different values of lambda ... Does this mean five separate configurations with $\lambda$ set to 0.5, 0.75, 1.0, 1.25, and 1.5 respectively? The wording is a bit unclear."
>
> **A4 & A5.** Thanks for your question. We will clarify this point in the next version of our paper. The following table shows $\lambda$ for each experiment:
>
> |  | Burgers' Equation | Schrödinger Equation | Navier-Stokes Equations |
> |---|---|---|---|
> | ABL-PDE (w/o FEM) | 0.05 | 0.05 | 1 |
> | ABL-PDE | 5 | 0.05 | 200 |

---

> > ### Comment · Reviewer_dxqN · 2025-08-04
> >
> > I appreciate the authors' clarifications and the additional experimental results. I have a few follow-up questions:
> >
> > - (For Q1) Could the authors elaborate on the reason for setting the margin threshold to 1e-3? Specifically, is there a risk that this threshold could eliminate terms that are actually part of the ground truth? Since this is a key hyperparameter of the ABL-PDE method, could the authors clarify the relationship between its value and your model's performance or robustness?
> >
> > - (For Q1) I also have a follow-up regarding the L2RE metric. Suppose the ground truth expression is $2x_0+x_0\cdot x_1$, How would the L2RE (if applicable) be computed for the following recovered equations: (1) $2x_0+x_0\cdot x_1^2$ (2) $2x_0+2x_0\cdot x_1+x_0\cdot x_1^2$ (3) $x_0\cdot x_1$?
> >
> > - (For Q1) Would the authors consider reporting the success rate (recovery rate) alongside L2RE as a more interpretable measure of correctness? Many related works include it, and some even treat it as the primary evaluation metric.
> >
> > - (For Q2) Thank the authors for the effort in adapting ABL-PDE to the benchmarks used by SGA-PDE and SPL. Based on the reported results, could the authors provide a summary of the key benefits or trade-offs of ABL-PDE compared to these recent baselines? For example, under what conditions would ABL-PDE be preferable over SGA-PDE or SPL?
> >
> > - (For Q4&5) It appears that the values of $\lambda$ differ between ABL-PDE and ABL-PDE (w/o FEM), and across different equations. Are these choices primarily driven by performance tuning? If so, when applying ABL-PDE to an unseen PDE system, how should one determine a suitable value of $\lambda$? If these values are not fixed in advance and require tuning, it may be helpful to propose a deterministic procedure or report the experiment results under a consistent $\lambda$.

---

> > > ### Author Response · Authors · 2025-08-05
> > >
> > > Thank you very much for your questions and suggestions! We will revise our paper accordingly. Below, we answer your questions in detail.
> > >
> > > **Q1'.** Concerns regarding margin threshold and evaluation metric.
> > >
> > > **A1'.** Empirically, we have found that during consistency optimization, it is easier to eliminate redundant terms than it is to recover omitted ones. For this reason, we select a small margin threshold with the goal of including potentially relevant terms during the initial execution of the MSS algorithm. You are absolutely correct that an unsuitable margin threshold could eliminate terms that are actually part of the ground truth. However, our experimental results show that $1 \times 10^{-3}$ is a quite robust choice. We commit to adding results under different margin thresholds to the next version of the paper, along with the success rate as a new metric.
> > >
> > > **Q2'.** Concerns regarding L2RE.
> > >
> > > **A2'.** We sincerely apologize for the confusion. We agree that changing the metric from L2RE to the **sum of relative errors** would be clearer.
> > >
> > > Here are the calculation results for the scenarios you mentioned, given the ground truth equation of $2x_0 + x_0 x_1$:
> > >
> > > + For $2x_0 + x_0 x_1^2$:
> > > $ |\frac{2 - 2}{2}| + |\frac{0 - 1}{1}| = 1 $
> > > + For $2x_0 + 2x_0 x_1+x_0 x_1^2$:
> > > $ |\frac{2 - 2}{2}| + |\frac{2 - 1}{1}| = 1 $
> > > + For $x_0 x_1$:
> > > $ |\frac{0 - 2}{2}| + |\frac{1 - 1}{1}| = 1 $
> > >
> > > **Q3'.** Comparison of ABL-PDE with SGA-PDE and SPL.
> > >
> > > **A3'.** We appreciate the opportunity to clarify the key distinctions between our method and other influential studies. The primary differences are rooted in two fundamental aspects: the method of derivative computation and the approach to equation term representation.
> > >
> > > + **Derivative Computation Method.**
> > >
> > >     The approaches to derivative computation represent a fundamental divergence. SGA-PDE and SPL both rely on **numerical differentiation**. This reliance necessitates a complete, structured mesh grid and makes the methods inherently sensitive to measurement noise, which is why SPL employs a Savitzky–Golay filter for data smoothing. In contrast, ABL-PDE approximates the physical field with a neural network and calculates derivatives using **automatic differentiation**. This offers a significant advantage, as our method only requires arbitrarily distributed spatio-temporal data points, relaxing the stringent requirement for a structured grid. More crucially, the derivative approximation methods in SGA-PDE and SPL contain no learnable parameters, making it impossible for them to refine derivative estimates using the physical laws they discover. The ability to **leverage discovered physical information to refine derivative estimation** is a core contributor to the robustness and accuracy of ABL-PDE.
> > >
> > >     Of course, we acknowledge that a neural network approximation is not always superior. For instance, with very sparse data, approximation methods with stronger inductive biases, such as Gaussian Processes or Fourier series, may yield better derivative estimates. A key strength of our framework is its modularity; it can seamlessly integrate these alternative function approximators. The learnable parameters of these approximators (e.g., length scales in Gaussian Processes) can then be optimized via the consistency optimization process.
> > >
> > > + **Equation Term Representation.**
> > >
> > >     The methods also differ significantly in their representation of equation terms. SGA-PDE uses a binary tree, while SPL employs a context-free grammar. ABL-PDE, on the other hand, utilizes a **FOL knowledge base**. While all three representations are highly expressive, our approach provides a unique advantage: it allows for the easy and explicit incorporation of **expert knowledge** through **integrity constraints**. This mechanism enables the encoding of rich, high-level structural rules and physical priors, which dramatically prunes the search space.
> > >
> > > **Q4'.** Concerns regarding the choice of $\lambda$.
> > >
> > > **A4'.** Thank you for your insightful question. Please note that we do not select the hyperparameter $\lambda$ based on the performance of the equation discovery. Instead, as mentioned in Appendix F.3, we choose $\lambda$ based on the fitting error of the physical field on the validation set.
> > >
> > > We do not perform a fine-grained search for $\lambda$. A value is considered acceptable as long as the fitting error on the validation set is close to or less than the pre-training error (in the presence of noise, it may not necessarily be less than the pre-training error). Furthermore, our experiments under different values of $\lambda$ have demonstrated the method's robustness to its selection.
> > >
> > > Systematically providing a definitive algorithm for choosing $\lambda$ is indeed a challenging problem in Physics-Informed machine learning. Empirically, a common practice is to perform a coarse grid search over several orders of magnitude for $\lambda$ and select the one that yields the best performance on the validation set.

---

> > > > ### Comment · Reviewer_dxqN · 2025-08-07
> > > >
> > > > Thank you to the authors for the clarifications regarding the margin threshold, L2RE metric, and the selection of $\lambda$. However, several key concerns remain:
> > > >
> > > > (1) Hyperparameter sensitivity: margin threshold and λ
> > > > While I appreciate the explanation of the design intuition and the mention of future additions to the paper, I believe an ablation study remains necessary to validate how the performance of the method is affected by the choices of these hyperparameters. In particular, as shown in the original rebuttal (A4 & A5), the selected $\lambda$ values across experiments vary significantly (e.g., 0.05, 1, 200), suggesting a lack of consistency or robustness.
> > > > The proposed solution - performing a coarse grid search over several orders of magnitude for each equation - is not a practical or scalable strategy, especially when no prior knowledge of the target equation is available. In real-world scenarios, users typically will not have access to ground-truth structures or ideal hyperparameter ranges. A more practical and compelling demonstration would be to show that the method is reasonably robust to variations in these hyperparameters, ideally with recommendations for default values that generalize across tasks.
> > > >
> > > > (2) Use of the L2RE metric
> > > > I appreciate the authors' acknowledgment that better alternatives exist. As currently used, L2RE fails to reflect some key aspects of equation recovery accuracy. Specifically:
> > > >
> > > > - It does not properly account for term importance, particularly when non-dominant terms with large coefficients, are missing or underestimated.
> > > >
> > > > - It does not penalize structural errors, such as the number and complexity of incorrect extra terms. For example, under L2RE, including 10 incorrect terms can be treated similarly to including just 1.
> > > > Replacing L2RE with a more informative metric - or providing results under alternative metrics - would significantly improve the evaluation.
> > > >
> > > > (3) Comparison with baselines under noise
> > > > While the theoretical advantages of ABL-PDE over SGA-PDE and SPL in noisy settings are clearly articulated, I find that the experimental validation in A2 does not yet reflect these benefits convincingly. In particular, the performance gains on noisy datasets are not clearly evident.
> > > > I appreciate the additional experiments, but I remain unconvinced that the proposed method provides a meaningful improvement over baselines in noisy scenarios. Stronger empirical evidence is needed to support the claimed robustness advantage.
> > > >
> > > > Given the above, I will maintain my original score for now, while remaining open to revisiting it during the post-discussion phase in coordination with the other reviewers.

---

> > > > > ### Author Response · Authors · 2025-08-08
> > > > >
> > > > > Thank you very much for your detailed review. We address your concerns below.
> > > > >
> > > > > **Q1''.** Concern regarding hyperparameter selection.
> > > > >
> > > > > **A1''.** As explained in our previous response (A1'), the chosen margin threshold consistently works across datasets, demonstrating its effectiveness. We commit to add results from different thresholds to our revision.
> > > > >
> > > > > Regarding your concern about the selection of $\lambda$, we sincerely apologize for not providing the experimental data previously due to the tight rebuttal schedule. We have now conducted experiments for $\lambda$ values spanning several orders of magnitude under the condition of 10% noise for the Burgers' Equation. The results are shown in the table below ($\lambda$ used in our paper was 5):
> > > > >
> > > > > | Method | $\lambda$ | $u \partial_x u$ | $\partial_{xx} u$ | Val Err(%) |
> > > > > |---|---|---|---|---|
> > > > > | ABL-PDE(w/o CO)$^&dagger;$ | - | -0.8527 | 0.0035 | 1.88 |
> > > > > | ABL-PDE | 0.5 | -0.9830 | 0.003282 | 0.80 |
> > > > > | | 5 | -0.9783 | 0.003201 | 0.86 |
> > > > > | | 50 | -0.9689 | 0.006270 | 4.04 |
> > > > > | | 500 | -0.9087 | 0.01098 | 8.77 |
> > > > > | Ground Truth | - | -1.0 | 0.003183 | - |
> > > > >
> > > > > Here, the "Validation Error" is L2 relative error for the physical field on the validation set. Only ABL-PDE (w/o CO) includes two redundant terms, $u^3$ and $u$, with coefficients of 0.0175 and -0.0186, respectively.
> > > > >
> > > > > As shown in the table, $\lambda$ values yielding better validation performance also produce more accurate coefficients. Therefore, selecting $\lambda$ based on the validation set is a reasonable and practical strategy.
> > > > >
> > > > > Regarding the default values, based on our experience, one could try values such as 0.1, 1, 10, and 100, or adopt the rescaling trick mentioned in A2.
> > > > >
> > > > > **Q2''.** Concern regarding the L2RE metric.
> > > > >
> > > > > **A2''.** Thanks for your suggestion.
> > > > > + We highly agree with your consideration regarding term importance, which is precisely the reason we chose the L2RE metric. The relationship between the magnitude of a term's coefficient and its physical importance is often not straightforward. For instance, in the case of the Burgers' equation discussed in our paper, the coefficient of $\partial_{xx} u$ is merely 0.00318, which is two orders of magnitude smaller than the coefficient of $u \partial_x u$ (-1.0). However, the equation including $\partial_{xx} u$ is the **viscous** Burgers' equation, while the one without it is the **inviscid** Burgers' equation, and the properties of their respective solutions differ significantly. Based on the consideration that a small coefficient does not necessarily imply unimportance, we opted not to use the RMSE metric (as used in D-CIPHER[1]), because the errors from large coefficients would dominate those from small ones. Instead, we calculate the sum of the relative errors for each coefficient, which is precisely L2RE.
> > > > >
> > > > > + We noted that SPL provides the number of redundant terms in the main text, with the specific forms detailed in the appendix. Following a similar principle due to space constraints, we commit to add the specific forms of all discovered equations to the appendix.
> > > > >
> > > > > **Q3''.** Concern regarding noise robustness.
> > > > >
> > > > > **A3''.** We would like to clarify that SGA-PDE does not involve noisy data in their experiments. SGA-PDE calculates derivatives via numerical differentiation on a full mesh grid. In contrast, our method operates on sparse data. For the Burgers' equation, we used 1e4 points (around 25% of the grid data); for the Chafee-Infante equation, 3e4 points (around 50% of the grid data); and for the KdV equation, 2e4 points (around 20% of the grid data). The number of points was selected empirically and was not tuned based on experimental results.
> > > > >
> > > > > In response to your concern about hyperparameter selection, we employed a unified loss rescaling strategy.  It would be straightforward to achieve a better coefficient estimation performance by fine-tuning $\lambda$, as the quality of the coefficients can be clearly reflected by a noise-free validation set.
> > > > >
> > > > > The experiments in SPL include 5% white noise. Since both our method and SPL successfully identified the correct equations, we compared their L2RE performance. The results are as follows:
> > > > > * **SPL:** 8.78%
> > > > > * **ABL-PDE:** **3.53%**
> > > > >
> > > > > We commit to adding a comparison and discussion with SGA-PDE and SPL to the next version of the paper.
> > > > >
> > > > > As mentioned by Reviewer B51F, D-CIPHER[1] is a noise-robust method. We report the comparison results below:
> > > > >
> > > > > | | Burgers' Equation | | | Heat Equation | | |
> > > > > |---|---|---|---|---|---|---|
> > > > > | Noise Level | 0 | 5% | 10% | 0 | 5% | 10% |
> > > > > | D-CIPHER | 1.61e-2 | 8.13e-2 | 1.49e-1 | 4.37e-4 | 1.87e-2 | 2.62e-2 |
> > > > > | ABL-PDE | **1.77e-3** | **4.17e-3** | **7.11e-3** | **4.04e-4** | **5.03e-5** | **1.17e-3** |
> > > > >
> > > > > To align with the original paper, the evaluation metric is RMSE from D-CIPHER, and the results demonstrate our robustness against noise.
> > > > >
> > > > > Reference
> > > > >
> > > > > [1] D-CIPHER: discovery of closed-form partial differential equations. NeurIPS, 2023.

---

### Official Review · Reviewer_htDg · 2025-07-02

**Clarity:** 3
**Significance:** 3
**Originality:** 3
**Rating:** 4
**Confidence:** 3

**Summary:**

The idea of this paper is to integrate a neural network for fitting the solution field (and estimating derivatives) with a first-order logic (FOL) knowledge base that generates canonical expression terms via abductive logic programming.

**Questions:**

can you add add a synthetic example where the underlying PDE has unknown nonlinear parameter functions to see how the current framework fails or could be extended.

**Ethical Concerns:**

["NO or VERY MINOR ethics concerns only"]

**Limitations:**

They test only on well-known equations (Burgers, Schrödinger, Navier-Stokes in 2D).
If the initial pre-training fails (very sparse or chaotic data), the abductive loop may not recover.

**Paper Formatting Concerns:**

Slightly clearer diagrams of the FOL grammar rules vs canonical form (the BNF could be summarized more visually).

**Quality:**

3

**Strengths And Weaknesses:**

Strength:  Combining FOL abductive learning to systematically constrain symbolic PDE search is quite original.
Weaknesses
While the canonical forms reduce combinatorial explosion, the abductive reasoning and MSS steps could still become heavy for large systems. It would help to see timings or scaling on more complex multi-physics PDEs or with >2 dependent variables.
Though standard PDEs were used, the inclusion of a more realistic engineering problem (e.g., turbulent flow, reaction-diffusion with unknown kinetics) could strengthen the practical impact.

---

> ### Author Rebuttal · Authors · 2025-07-29
>
> We thank Reviewer htDg for insightful review and appreciation on the originality of our work. Below, we answer your questions in detail.
>
> ---
>
> **Q1.** "can you add add a synthetic example where the underlying PDE has unknown nonlinear parameter functions to see how the current framework fails or could be extended."
>
> **A1.**
> Thanks for your question. As mentioned in the conclusion, the limitation of ABL-PDE lies in its inability to handle unknown parameters within nonlinear terms. For example, $\theta$ in $\frac{1}{\theta \cdot u + v}$ is a typical case.
> It should be noted that introducing such terms into the hypothesis space of our knowledge base is straightforward; the main challenge lies in computing the consistency measure, particularly in the Monotone Subset Selection (MSS) step. In our work, the minimal computational unit of MSS involves solving for the linear combination coefficients $\xi$ given a set of expression terms. This is a least squares problem with a closed-form solution, allowing for efficient computation. However, if unknown parameters are introduced within nonlinear terms, the minimal computational unit of MSS becomes an optimization problem that may lack a closed-form solution and can even be non-convex, resulting in significantly increased computational cost. This is precisely why we mentioned in our paper the necessity to incorporate advanced techniques for joint parameter optimization for future work.
>
> ---
>
> **Q2.** "Though standard PDEs were used, the inclusion of a more realistic engineering problem (e.g., turbulent flow, reaction-diffusion with unknown kinetics) could strengthen the practical impact."
>
> **A2.**
> Thanks for your suggestion. We highly agree that testing our method in more complex scenarios would better demonstrate its practical applicability.
> However, fitting unknown and complex kinetics often involves learning parameters within nonlinear terms.
> As we mentioned in A1, learning these parameters significantly increases computational cost and requires more advanced optimization techniques, which we consider as future work.
>
> ---
>
> **Q3.** "If the initial pre-training fails (very sparse or chaotic data), the abductive loop may not recover."
>
> **A3.** Thanks for your question. We acknowledge that such scenarios present significant challenges, and this is indeed an inherent difficulty in the broader problem of scientific discovery from data.
>
> In situations with very sparse data, where a dense grid is unavailable for traditional numerical differentiation, employing neural networks for function approximation is a widely adopted technique [1-2]. Our bidirectional loop acts as an enhancement to these data-driven approximation methods. It allows us to iteratively refine the model's understanding of the physical field by leveraging discovered physical insights, even if initial approximations are coarse due to data scarcity.
>
> For chaotic systems, methods with stronger inductive biases or prior knowledge—such as Gaussian Processes, Fourier series, or Chebyshev polynomials—might indeed offer better performance than a generic neural network. Fortunately, our framework is highly modular and flexible. As long as these alternative models incorporate learnable parameters (e.g., length scales and signal variance for Gaussian Processes), they can seamlessly substitute the "machine learning" part depicted in Figure 2 of our paper. This adaptability allows our overall system to leverage the strengths of different modeling paradigms, tailoring the data approximation component to the specific characteristics of the physical system.
>
> References:
>
> [1] DLGA-PDE: Discovery of PDEs with incomplete candidate library via combination of deep learning and genetic algorithm. Journal of Computational Physics, 2020.
>
> [2] Deep-learning based discovery of partial differential equations in integral form from sparse and noisy data. Journal of Computational Physics, 2021.

---

### Official Review · Reviewer_DouY · 2025-07-04

**Clarity:** 2
**Significance:** 2
**Originality:** 3
**Rating:** 5
**Confidence:** 3

**Summary:**

The authors present a new approach for automatic discovery of partial differential equations from data using neural network learning and first order logic based reasoning. Current approaches for partial equation discovery suffers from several issues including - large hypothesis space which makes it challenging to search for the correct equation, inaccurate derivative estimation during expression evaluation as only discrete measurements are available. To address these issues, the authors propose to use first-order-logic-based reasoning to constrain the hypothesis space of possible expressions/equations, and neural network based derivative estimation for expression evaluation. The authors’ framework also contains an optimization module to ensure consistency between these two modules. Experimental results are provided to demonstrate the efficacy of the proposed framework.

**Questions:**

Mentioned in the strength and weakness section

**Ethical Concerns:**

["NO or VERY MINOR ethics concerns only"]

**Final Justification:**

I had questions about the manuscript, the authors clarified and provided more details in their response. I have also read other reviewer comments, and I think most of the major concerns have been addressed by the authors. Therefore, I will suggest accept for this paper.

**Limitations:**

Please see the strength and weakness section

**Quality:**

3

**Strengths And Weaknesses:**

Strength:
The authors presented a good introduction and motivation for their work. Related works are discussed along with their limitations. Algorithm for Monotone subset selection is presented in detail. Theoretical analysis is presented. Experimental results are organized with key questions mentioned at the beginning. Ablation study is provided to show the importance of different modules of the framework.

Weakness:
1. The authors may consider adding the following details to their manuscript:
2. Why is it necessary to have a consistency maximization process? There can be one/two sentences between line 54-62 describing the intuition behind it.
3. In the contribution, the authors mention bidirectional loop, however symbolic regressions method like genetic programming also has this kind of iterative loop structure, how is the authors’ method different from that?
4. What is the input/output of the neural network used for derivative estimation? There are many methods for approximating derivatives and the authors mention their limitations in the introduction. How does having a neural network mitigate those issues?
5. Can the authors give an example of how abduction works as indicated in section 4.1, step 2. It is currently hard to understand how abduction relates the number of operators to expand the library of candidates.
6. Table 1 integrity constraints are hard to follow at present, despite the authors giving details. Can you please elaborate more on one constraint, maybe C1 as to why this will give a unique order.

---

> ### Author Rebuttal · Authors · 2025-07-29
>
> We thank Reviewer DouY for detailed review and suggestions on the clarity of our paper. We will revise the paper accordingly. Below, we answer your questions in detail.
>
> ---
>
> **Q1.** "Why is it necessary to have a consistency maximization process?"
>
> **A1.**
> Thanks for your question and suggestion. The intuition behind the consistency maximization process is to bridge the gap between the **continuous, differentiable machine learning component** and the **discrete, non-differentiable logical reasoning component**, as illustrated in Figure 2 of the main text.
>
> We can train the neural network with gradient descent to approximate the physical field, however, we cannot calculate the gradient of the equation structure with respect to the neural network output.
> Thus, we requires a calculable criterion to construct a connection between the equation structure and the neural network, such that the change of neural network parameters can quantitavely affect the equation structure.
> In simple words, the consistency measure makes the equation structure an implicit function of the neural network parameters, and we optimize the neural network and determine the equation structure jointly by maximizing the consistency measure.
>
> ---
>
> **Q2.** "In the contribution, the authors mention bidirectional loop, however symbolic regressions method like genetic programming also has this kind of iterative loop structure, how is the authors’ method different from that?"
>
> **A2.**
> Thanks for your question. We highly agree with you that many symbolic regression methods, such as those based on genetic programming, also employ iterative loop structures. However, the fundamental distinction of our method lies in the **nature and purpose of its bidirectional loop**.
>
> Specifically, our bidirectional loop operates between **neural network training** and **equation structure identification**. This contrasts sharply with the iterative loop structures found in other methods, which are confined solely to the realm of equation structure identification. As we mentioned in lines 51-53 of our paper, other approaches ignore the potential of using discovered physical information to enhance the model's ability to fit the physical field.
>
> This distinction applies to genetic programming-based methods like DLGA-PDE [1], DL-PDE [2], and SGA-PDE [3], as well as sparse regression-based methods like WeakSINDy [4] and D-CIPHER [5]. To the best of our knowledge, our method is the **first** to establish a mutually beneficial loop between model training and equation structure identification.
>
> References:
>
> [1] DLGA-PDE: Discovery of PDEs with incomplete candidate library via combination of deep learning and genetic algorithm. Journal of Computational Physics, 2020.
>
> [2] Deep-learning based discovery of partial differential equations in integral form from sparse and noisy data. Journal of Computational Physics, 2021.
>
> [3] Symbolic genetic algorithm for discovering open-form partial differential equations (SGA-PDE). Physical Review Research, 2022.
>
> [4] Weak SINDy for partial differential equations. Journal of Computational Physics, 2021.
>
> [5] D-CIPHER: discovery of closed-form partial differential equations. NeurIPS, 2023.
>
> ---
>
> **Q3.** "What is the input/output of the neural network used for derivative estimation? There are many methods for approximating derivatives and the authors mention their limitations in the introduction. How does having a neural network mitigate those issues?"
>
> **A3.** The input of the neural network is the spatial-temporal coordinate and the output is the predicted value of the physical field at that coordinate.
>
> We mitigate issues of derivative estimation through our **bidirectional enhancement loop** (i.e., consistency optimization). The core idea here is to leverage the discovered physical information to guide and improve the neural network's training. Even if this physical information is initially imprecise, the resulting improvements in the network's performance can, in turn, help refine the physical information itself, creating a mutually beneficial and iterative enhancement.
>
> Experimental results demonstrate the effectiveness of consistency optimization in improving the accuracy of derivative estimation.
> As discussed in our analysis of the Burgers' Equation experiment, the shock wave in the solution can only be captured by an extremely dense mesh that is impractical in most real-world applications. This limitation leads to suboptimal derivative estimation performance for the pre-trained network. Even when the form of the equation is known (as in PDERidge, Table 2), the coefficient estimation still have significant errors. **In contrast, our method achieves a 5x to 10x reduction in L2 relative error through consistency optimization**.
>
> ---
>
> **Q4 & Q5.** "Can the authors give an example of how abduction works as indicated in section 4.1, step 2. It is currently hard to understand how abduction relates the number of operators to expand the library of candidates. ... Table 1 integrity constraints are hard to follow at present, despite the authors giving details. Can you please elaborate more on one constraint, maybe C1 as to why this will give a unique order."
>
> **A4 & A5.** Thanks for your question. Due to space limitations, we use a simplified example—multiplication expressions with two dependent variables (u, v)—to detail the processes of abduction and the role of integrity constraints.
>
> First, we present a vanilla Prolog program without operator number limit and integrity constraints.
> ```prolog
> % Logical facts
> unit_expr(u).
> unit_expr(v).
>
> % Logical rules
> multiplication_expr(X) :- unit_expr(X).
> multiplication_expr(mul(X, Y)) :- unit_expr(X), multiplication_expr(Y).
> ```
> The semantic of the program is intuitive: a `multiplication_expr` is either a `unit_expr` or a `mul` with two arguments-the first being a `unit_expr`, and the second being a `multiplication_expr`.
> When querying `?- multiplication_expr(X).`, Prolog attempts to satisfy this goal by finding a value for `X` that fits the definition of a `multiplication_expr`. It finds answers using Selective Linear Definite clause resolution (SLD resolution), in other words, through **recursion** and **backtracking**. We explain the process in detail below.
> 1.  **First Rule: Base Cases**
>     * Prolog first tries this rule. It looks for simple "unit" expressions.
>     * It finds `unit_expr(u)` and reports `X = u`.
>     * Then it finds `unit_expr(v)` and reports `X = v`. These are the simplest, non-nested expressions.
>
> 2.  **Second Rule: Recursive Cases**
>     * After the simple cases, Prolog moves to the second rule to build more complex expressions.
>     * This rule says a `multiplication_expr` can be `mul(something, something_else)` where the first part is a `unit_expr` (like `u` or `v`), and the second part is *another* `multiplication_expr`.
>     * **How it works:**
>         * It picks a `unit_expr` for the first part.
>         * Then, for the second part, it **recursively** calls `multiplication_expr` again. This means it goes back to step 1 and 2 to find *another* expression for the second part.
>         * This recursion allows it to build nested structures like `mul(u, u)`, then `mul(u, v)`, `mul(u, mul(u, u))`, and so on.
>         * **Backtracking:** If Prolog hits a dead end or you ask for more solutions (by typing `;`), it backtracks. This means it goes back to the last choice it made and tries a different option to find more combinations and deeper nested structures.
>
> **Operator Number Limit**
>
> One issue with the above program is that it will generate endlessly nested structures like `mul(u, mul(u, mul(u, mul(u, ...))))`, and another subtle issue is that the number of `v` will never be greater than 1. By introducing operator number limit, we solve both issues.
>
> ```prolog
> unit_expr(u).
> unit_expr(v).
>
> multiplication_expr(X, Op_limit) :- 0 =< Op_limit, unit_expr(X).
> multiplication_expr(mul(X, Y), Op_limit) :- 1 =< Op_limit, unit_expr(X),
>                                             multiplication_expr(Y, Op_limit - 1).
> ```
>
> The only difference is that we add a new argument `Op_limit` to the `multiplication_expr` rule, which is the number of operators allowed in the expression.
> When querying `?- multiplication_expr(X, 1).`, only `u`, `v`, `mul(u, u)`, and `mul(u, v)`, `mul(v, u)`, `mul(v, v)` will be generated.
>
> **Integrity Constraints**
>
> Since we do not want to both generate `mul(u, v)` and `mul(v, u)`, we introduce the following integrity constraints to the second rule.
>
> ```prolog
> multiplication_expr(mul(u, v), Op_limit) :- 1 =< Op_limit, unit_expr(u),
>                                             multiplication_expr(v, Op_limit - 1),
>                                             \+ integrity_constraint(mul(X, Y)).
> integrity_constraint(mul(X, Y)) :- Y = mul(Z, _) -> X @> Z ; X @> Y.
> ```
> The semantics of `\+` is "not," meaning that for a valid `mul(X, Y)`, the boolean value of `integrity_constraint(mul(X, Y))` should be `false`.
> The definition of `integrity_constraint(mul(X, Y))` uses an if-then-else structure (`Condition -> Then ; Else`), which evaluates to `true` if the former term is greater than the latter term according to the meaning of `@>`, a built-in predicate in Prolog.
> By introducing this integrity constraint, we can prevent the generation of terms like `mul(v, u)`.
>
> For a more comprehensive example of our logic-based knowledge base implementation, please refer to Appendix B.

---

> > ### Comment · Reviewer_DouY · 2025-08-09
> >
> > Thank you authors for your detailed response. I have now a better understanding of the paper, these details included in the authors response are crucial for readers' understanding. Thank you again for your response, I will update my score accordingly.

---

> > > ### Author Response · Authors · 2025-08-09
> > >
> > > Once again, thanks for your very constructive comments! We will improve the clarity in the revised version according to your suggestions. Thanks!

---

### Comment · Area_Chair_qD1j · 2025-08-07
**Please finalize discussion**

Dear reviewers,

The authors have replied extensively to all your reviews. If you have not done so, please carefully read through the author responses as well as the other reviews and engage in the discussion. The discussion period is nearing it's end.
Please note that the "Mandatory Acknowledgement" includes a statement that you engaged in the discussion.
How do the author replies and other reviews affect your view of the submission? Please adjust your score accordingly.

Best,
AC

---

### Note · Authors · 2025-08-14

Dear Reviewers,

We deeply appreciate the reviewers' positive feedback, especially for highlighting our work's novelty (by Reviewer htDg, Reviewer dxqN, Reviewer B51F), effectiveness (by Reviewer dxqN, Reviewer B51F), and well-structured presentation (by Reviewer DouY, Reviewer B51F).

We have provided point-by-point responses to all concerns raised. To further polish and enhance our manuscript, we are fully committed to incorporating your suggestions in the next version.

Regarding the remaining concerns from Reviewer dxqN, we commit to making the following specific improvements in the next version of our paper:
1.  Incorporate the explicit forms of the discovered equations.
2.  Introduce the "recovery rate" as an additional evaluation metric.
3.  Incorporate additional experiments on the impact of the margin threshold and an even wider range of $\lambda$ values on performance.
4.  Incorporate the comparative experiments (A2) and discussion (A3') with SGA-PDE and SPL.

Once again, we thank all of you for your time and helpful suggestions. We remain confident in our work's contribution to the community, particularly in how our proposed bidirectional loop enhances the robustness of PDE discovery and how the FOL knowledge base effectively constrains the search space while integrating expert knowledge.

Best Regards,

The Authors

---

### Decision · Program_Chairs · 2025-09-17

**Decision:**

Accept (poster)

**Comment:**

This submission combines deep learning, a first-order logic knowledge base and consistency optimization to discover symbolic PDEs. The proposed bidirectional loop appears to be novel and the experimental results across benchmarks and noise levels are quite strong. The authors provided an extensive rebuttal and engaged extensively in the discussion throughout which they managed to mostly clarify the theoretical soundness and convinced most reviewers of the originality and applicability of the work. One reviewer remained slightly unconvinced but does not strongly oppose acceptance. Given the convincing clarifications and broad agreement among reviewers, I recommend accept as a poster.